# Polyclonal antibody responses to HIV Env immunogens resolved using cryoEM

Aleksandar Antanasijevic[1,2], Leigh M. Sewall[1,2], Christopher A. Cottrell [1,2], Diane G. Carnathan[3], Luis E. Jimenez[3], Julia T. Ngo[3], Jennifer B. Silverman [3], Bettina Groschel[4], Erik Georgeson [4], Jinal Bhiman[4], Raiza Bastidas[4], Celia LaBranche [5], Joel D. Allen [6], Jeffrey Copps [1], Hailee R. Perrett [1], Kimmo Rantalainen [4], Fabien Cannac[1], Yuhe R. Yang[1], Alba Torrents de la Peña[1], Rebeca Froes Rocha[1], Zachary T. Berndsen[1], David Baker [7,8], Neil P. King [7], Rogier W. Sanders[9,10], John P. Moore [10], Shane Crotty [11], Max Crispin [6], David C. Montefiori[5], Dennis R. Burton [2,4], William R. Schief [2,4], Guido Silvestri[3] & Andrew B. Ward [1,2✉]

Engineered ectodomain trimer immunogens based on BG505 envelope glycoprotein are widely utilized as components of HIV vaccine development platforms. In this study, we used rhesus macaques to evaluate the immunogenicity of several stabilized BG505 SOSIP constructs both as free trimers and presented on a nanoparticle. We applied a cryoEM-based method for high-resolution mapping of polyclonal antibody responses elicited in immunized animals (cryoEMPEM). Mutational analysis coupled with neutralization assays were used to probe the neutralization potential at each epitope. We demonstrate that cryoEMPEM data can be used for rapid, high-resolution analysis of polyclonal antibody responses without the need for monoclonal antibody isolation. This approach allowed to resolve structurally distinct classes of antibodies that bind overlapping sites. In addition to comprehensive mapping of commonly targeted neutralizing and non-neutralizing epitopes in BG505 SOSIP immunogens, our analysis revealed that epitopes comprising engineered stabilizing mutations and of partially occupied glycosylation sites can be immunogenic.

[1] Department of Integrative, Structural and Computational Biology, The Scripps Research Institute, La Jolla, CA, USA. [2] International AIDS Vaccine Initiative Neutralizing Antibody Center, the Collaboration for AIDS Vaccine Discovery (CAVD) and Scripps Consortium for HIV/AIDS Vaccine Development (CHAVD), The Scripps Research Institute, La Jolla, CA, USA. [3] Division of Microbiology and Immunology, Yerkes National Primate Research Center, Emory University, Atlanta, GA, USA. [4] Department of Immunology and Microbiology, The Scripps Research Institute, La Jolla, CA, USA. [5] Department of Surgery, Duke University Medical Center, Durham, NC, USA. [6] School of Biological Sciences, University of Southampton, Southampton, UK. [7] Institute for Protein Design, Department of Biochemistry, University of Washington, Seattle, WA, USA. [8] Howard Hughes Medical Institute, Chevy Chase, MD, USA. [9] Academic Medical Center (AMC), University of Amsterdam, Amsterdam, Netherlands. [10] Weill Cornell Medicine, Cornell University, New York, NY, USA. [11] La Jolla Institute for Immunology, La Jolla, CA, USA. ✉email: andrew@scripps.edu

Envelope glycoprotein (Env) trimers derived from the BG505 genotype are the basis of many current HIV vaccine development efforts[1–9]. When expressed as ectodomain constructs stabilized with SOSIP mutations, trimers based on this clade A sequence can be readily produced at high yields while preserving the native-like, pre-fusion conformation targeted by known broadly neutralizing antibodies (bnAbs)[1]. Importantly, stabilization also reduces the exposure of epitopes for non-nAbs. BG505 SOSIP has been subject to many immunogenicity studies and is under evaluation in human clinical trials (ClinicalTrials.gov Identifiers: NCT03699241, NCT04177355, and NCT04224701).

Since its discovery, efforts have been made to improve the original BG505 SOSIP design by incorporating additional mutations. These aim to maximize the in vitro and in vivo stability of SOSIP trimers and increase expression levels[10–13]. These constructs have also been endowed with germline-targeting mutations, resulting in extensive engineering at the sequence level[2,3,11]. In parallel, various nanoparticle platforms have been developed for trimer presentation to enhance in vivo trafficking properties and interaction with B-cells[6,14–19].

While BG505 SOSIP trimers have consistently elicited autologous NAb responses in rabbit and macaque animal models[8,10,20,21], that were shown to be protective in macaques[22], they have failed to induce broadly neutralizing responses[23]. This is partly due to limited accessibility and immunoquiescence of bnAb epitopes, caused by extensive glycan shielding and sequestration of functionally essential protein domains in the quaternary structure[24]. Conversely, the antibody response is redirected towards the readily accessible, and, in some cases immunodominant, epitopes that are typically strain-specific. Comprehensive mapping of such "off-target" epitopes in BG505 SOSIP and structural characterization of elicited antibodies will provide essential information for engineering the next generation of BG505-based immunogens with enhanced on-target reactivity. The standard approach would be to apply B-cell sorting to isolate representative monoclonal antibodies (mAbs) from different polyclonal families and perform structural characterization of each mAb. This approach, while valuable, is laborious in nature and impractical for routine large-scale applications.

Herein, we introduce cryoelectron microscopy-based polyclonal epitope mapping (cryoEMPEM): a method for rapid, high-resolution structural characterization of antibody-antigen complexes without the need for mAb isolation. We applied cryoEMPEM in combination with ELISA and pseudovirus inhibition assays to characterize the polyclonal antibody (pAb) responses elicited by BG505 SOSIP immunogens in rhesus macaques. The main goal was to acquire a detailed, molecular-level understanding of the immunogenic landscape of stabilized BG505 SOSIP constructs as soluble trimers and presented on a nanoparticle. Twenty-one high-resolution maps of trimer immune complexes with polyclonal Fabs provided detailed insights to the nature of antibody responses at eight unique epitope clusters predominantly targeted in BG505 SOSIP immunogens.

## Results

**Immunogenicity of stabilized BG505 SOSIP trimer immunogens.** Two BG505 SOSIP trimer antigens bearing different sets of stabilizing mutations, BG505 SOSIP MD39 and BG505 SOSIP. v5.2 N241/N289 were evaluated in this study (Supplementary Fig. 1)[10,11]. The BG505 SOSIP.v5.2 N241/N289 construct included glycosylation sites at positions N241 and N289 to reduce access to the immunodominant glycan hole that exists in BG505 but is absent in the majority of HIV strains[20,25].

Two groups of 6 Rhesus macaques were injected with 100 µg of BG505 SOSIP MD39 (Grp 1) or BG505 SOSIP.v5.2 N241/N289 (Grp 2) antigens at 4 different time points (Fig. 1A). Each animal received the same antigen for all four immunizations. Serum/plasma samples were collected at 2-week intervals to monitor the development of antigen-specific antibodies by ELISA and NAbs using the TZM-bl pseudovirus inhibition assay. All animals elicited antigen-specific responses that were within ~1 order of magnitude for animals in each group at the corresponding time points (Fig. 1B, Supplementary Table 1). Grp 1 animals had lower average binding titers compared to Grp 2 after the first two immunizations, but this difference ceased after the third and fourth immunizations.

NAbs at serum neutralization titers ($ID_{50}$) above 40, were first observed after the second immunization (Fig. 1C, Supplementary Table 2). NAb titers showed substantial variability between animals; $ID_{50}$ values ranged from ~$10^1$ to $10^4$ at week 38. Three animals in Grp 1 and two animals in Grp 2 failed to develop detectable levels of NAbs after four immunizations. Altogether, through ELISA and pseudovirus inhibition assays, we detected consistent elicitation of antigen-specific antibodies in all animals, while the levels of NAbs varied significantly between animals. Overall, we observed a good correlation of both, binding and NAb titers to previously published BG505 SOSIP immunization experiments in rhesus macaque animal model[7,8,20,21,26]; although, the average titers were lower compared to studies in which optimized antigen delivery approaches were used (i.e., prolonged antigen release via osmotic pumps)[7,8].

We applied negative stain EMPEM (nsEMPEM) to identify the epitope specificities of pAbs elicited by animals in the two groups[27,28]. Immune complexes were prepared for imaging using either BG505 SOSIP MD39 or BG505 SOSIP.v5.2 N241/N289 with polyclonal Fabs obtained from plasma samples taken at week 26. The nsEMPEM analysis revealed that all animals developed antibody responses against the base of the trimer, a highly immunodominant epitope in SOSIP constructs (Fig. 1D and Supplementary Fig. 2)[29–31]. A C3/V5 response was present in three out of six animals in both groups, and it correlated well with autologous neutralization. Differences were also observed when comparing the vaccine-elicited antibody responses elicited in the two groups of animals. N241/N289 glycan hole responses were more prevalent in animals immunized with BG505 SOSIP MD39 (four animals in Grp 1 had this response but only one in Grp 2), suggesting that the presence of N241 and N289 glycans in the BG505 SOSIP.v5.2 N241/N289 immunogen suppressed the antibody response against this epitope. This is further supported by the results from Grp 3 of animals (see below), where we also used BG505 SOSIP with N241/N289 glycan knock-ins, and none of the animals have developed a detectable antibody response to the glycan hole. Overall, antibody responses against more diverse epitopes (e.g., N611 glycan epitope, fusion peptide, gp120-gp120 interface, and V1/V2/V3) were detected in animals from Grp 2 than in Grp 1. Antibodies targeting the gp120–gp120 interface, a new epitope, were present in all six animals from Grp 2 but absent in Grp 1 animals.

**CryoEMPEM analysis of the BG505-specific pAb responses.** Based on the nsEMPEM analysis, we selected polyclonal Fabs isolated from animals Rh.32034 (Grp 1), Rh.33104 (Grp 1) and Rh.33311 (Grp 2) for cryoEM studies. Antibody specificities detected in these three animals represented all unique epitope clusters (Fig. 1) except the V1/V2/V3 epitope, which was targeted only in animal Rh.CG41. Antibodies to this epitope were also detected in animals immunized with the nanoparticle

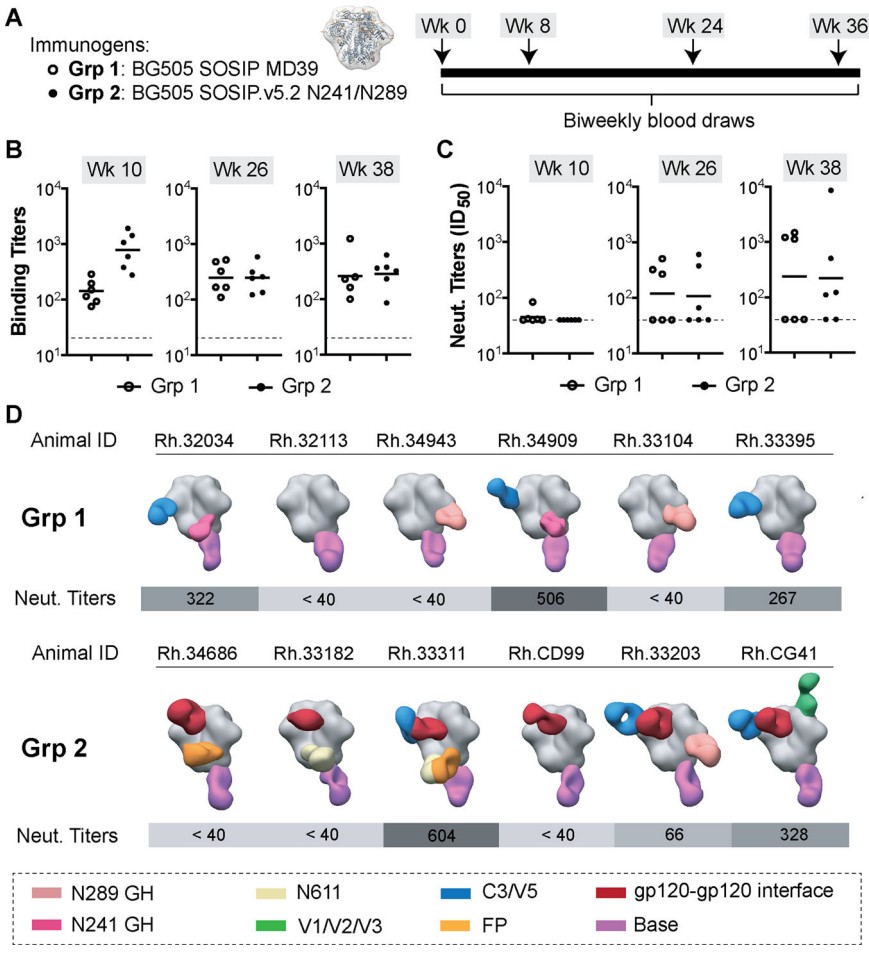

**Fig. 1 Immunization experiments with the stabilized BG505 SOSIP trimers. A** Immunogen (left) and immunization schedule (right). **B** ELISA binding titers (midpoint) and **C** neutralizing antibody titers (ID$_{50}$) for plasma samples collected at weeks 10, 26, and 38 (time points indicated above each graph; open circles—Grp 1; closed circles—Grp 2). Horizontal lines represent the geometric mean values at each time point (*n* = 6 animals in each group). ELISA and pseudovirus neutralization experiments were performed in triplicates (*n* = 3) for each serum sample at every time point and the mean value is plotted. **D** Composite figures from nsEMPEM analysis of polyclonal responses at week 26. Animal IDs and neutralizing antibody titers (at week 26) for corresponding animals are shown above and below each composite figure, respectively. A color-coding scheme for antibodies targeting different epitope clusters is shown at the bottom. BG505 SOSIP antigen is represented in gray.

immunogen (Grp 3, see below), and cryoEM characterization was instead performed on the samples from Grp 3.

Polyclonal immune complexes for cryoEMPEM analysis were prepared using a protocol equivalent to nsEMPEM, but with significantly higher starting amounts of both, antigen (~250 µg) and pAb (5–10 mg), necessary to achieve optimal particle density on cryoEM grids. SEC was applied to separate the polyclonal immune complexes from unbound Fabs (Supplementary Fig. 3). CryoEM grids were prepared and imaged as described in the Methods section, and data collection statistics are shown in Supplementary Table 3.

In the core of cryoEMPEM method is a data processing pipeline based on consecutive rounds of 3D classification with spherical masks positioned around the epitopes targeted by pAbs (Fig. 2A). This is done to gradually enrich the homogeneous subpopulations of immune complexes with shared structural features (i.e., presence/absence of pAbs and their relative orientation) and reduce the heterogeneity in the data caused by the differential content of pAbs. A detailed description of the approach is provided in the "Methods" section and illustrated in Supplementary Fig. 4.

From the three immune-complex samples, we reconstructed 16 high-resolution maps of structurally unique pAb classes (pAbC)

bound to the corresponding BG505 SOSIP antigen used in the immunization (Fig. 2B). Relatively large initial datasets (0.7–1.6 million clean particles after the symmetry-expansion step) enabled discrimination of Fabs bound to overlapping epitopes. This was particularly the case with base-targeting antibodies as they constitute the most prevalent type of antigen-specific antibodies (Fig. 2B, purple). Multiple pAbC classes targeting the N241/N289 glycan hole were also computationally isolated in the sample from animal Rh.33104, as well as C3/V5- and gp120–gp120 interface-directed antibodies in the sample from animal Rh.33311 (Fig. 2B).

The apparent resolutions of reconstructed maps ranged from 3.3 to 6.6 Å with an average of ~4.0 Å (Fig. 2B and Supplementary Fig. 5). In most cases, the part of the map corresponding to the trimer was resolved to a higher resolution than the Fab, as evident from the local resolution plots displayed in Supplementary Fig. 6. This is likely a consequence of the compositional and conformational heterogeneity stemming from the polyclonal nature of bound antibodies. Nevertheless, epitope–paratope interfaces were well-resolved in all EM maps, allowing for the identification of amino acids comprising the epitope for each pAb.

Atomic models of trimer-Fab immune complexes were relaxed into the reconstructed EM maps with apparent resolution ≤4.6 Å

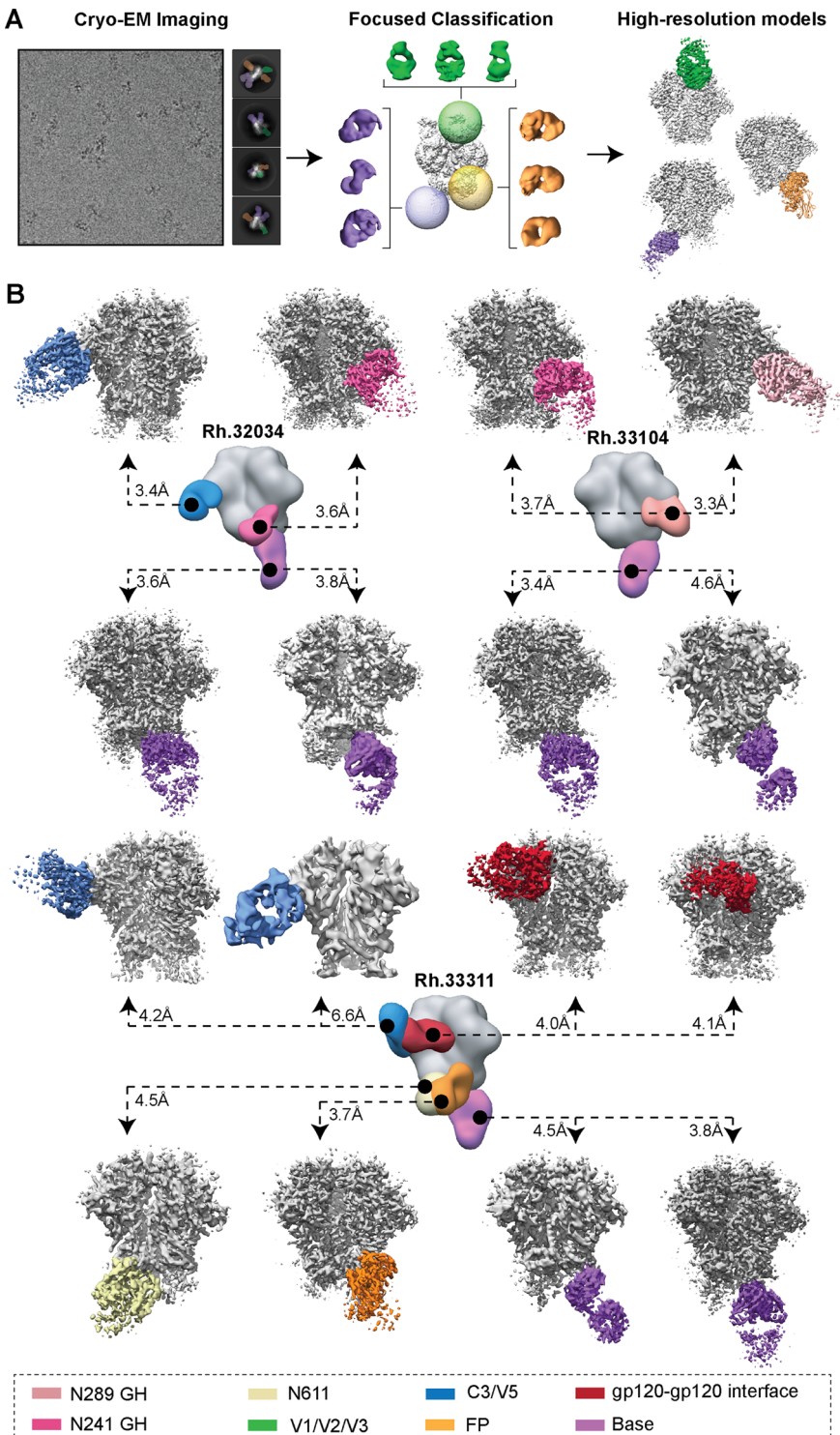

**Fig. 2 CryoEMPEM analysis of polyclonal antibodies elicited by BG505 SOSIP trimer immunogens. A** Schematic illustration of the data processing workflow of cryoEMPEM. **B** CryoEMPEM analysis of immune complexes generated using polyclonal Fabs isolated from plasma samples from animals Rh.32034 (top left), Rh.33104 (top right), and Rh.33311 (bottom). High-resolution trimer-Fab complexes featuring structurally unique antibody specificities detected in cryoEM datasets are shown in the corresponding panels. BG505 SOSIP antigen is represented in gray and Fab densities are colored according to the legend shown at the bottom. In the center of each panel is a composite figure from nsEMPEM. The apparent resolution of each reconstructed cryoEM map is indicated.

(15/16 maps met this criterion). Each Fab was represented as a poly-alanine backbone and comparison to published rhesus macaque Fab structures[32–34] was used to assign the heavy and light chains. We applied these models and heavy/light assignments to determine the role of specific complementarity determining regions (CDR) and framework regions (FR) in antigen recognition. Structure refinement information can be found in the Methods section and the relevant statistics are shown in Supplementary Table 4. Model-to-map fit for each complex structure is shown in Supplementary Fig. 7.

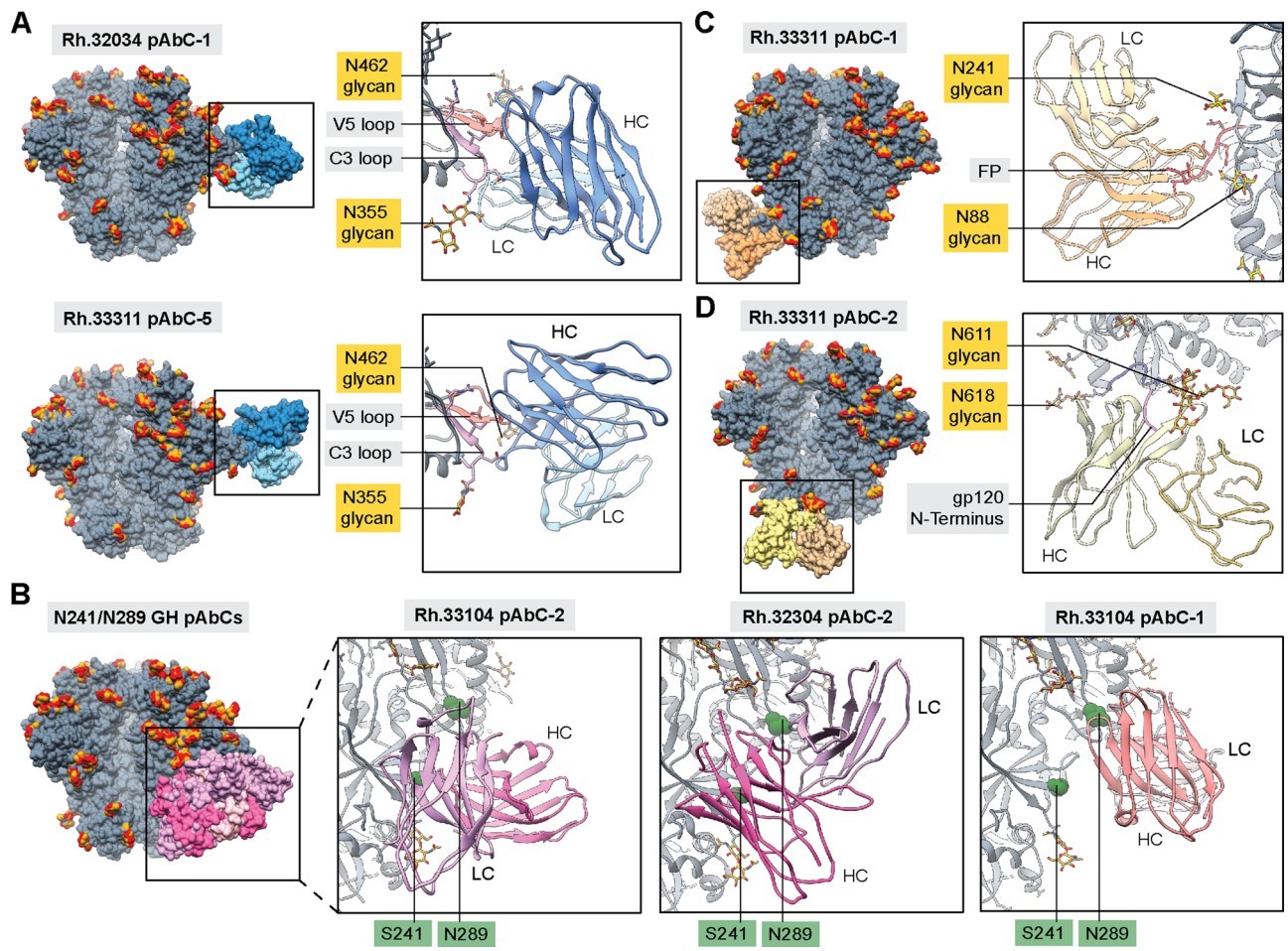

**Fig. 3 Structural analysis of neutralizing and potentially neutralizing antibody responses.** Structures of BG505 SOSIP antigens in complex with polyclonal antibodies targeting C3/V5 (**A**), N241/N289 glycan hole (**B**), fusion peptide (**C**), and N611 glycan epitopes (**D**). BG505 SOSIP trimers are shown in gray with N-linked glycans in golden yellow. Antibodies are colored using the same scheme as in Figs. 1 and 2. Inferred heavy and light chains for each antibody are represented in different shades and labeled in each panel (HC and LC, respectively). The most relevant epitope/paratope components are indicated in the supporting panels. Surface representation is used for the full immune complex (left side of each panel) while ribbon and stick representations are used for the display of epitope-paratope contacts (right side of each panel). In panel **B**, sphere representations are used for residues S241 and N289 to improve visibility.

For clarity, the analysis of pAbC structures is performed on a per-epitope basis.

**Antibodies targeting the C3/V5 epitope.** As described above, a correlation was observed between the elicitation of C3/V5-directed antibody responses and autologous neutralization (Fig. 1). CryoEMPEM analysis of polyclonal samples isolated from macaque Rh.32034 and Rh.33311 yielded three maps with pAbs targeting this site (Fig. 2B, Supplementary Fig. 7A). Analysis of the reconstructed models for Rh.32034 pAbC-1 and Rh.33311 pAbC-5 revealed that these two polyclonal antibodies targeted largely overlapping epitopes at the interface of C3 and V5 regions (residues 354–358 and 459–466, respectively) flanked by glycans at positions N355 and N462 (Fig. 3A). This epitope is not well-protected by the glycan shield and can be accessed from different angles. Indeed, the two reconstructed pAbCs displayed significantly different angles of approach and binding modes. In the case of Rh.33311 pAbC-5, the interaction is driven primarily by the heavy chain, while Rh.32034 pAbC-1 binds using both heavy and light chains. Notably, these are the first high-resolution structures of Env trimer complexes with antibodies targeting this epitope.

The neutralization potential of the C3/V5 epitope in BG505 has been reported previously in studies with a macaque, rabbit,

guinea pig, and mouse animal models[7,8,20,26,28,35–37]. Antibodies bound to this site can sterically block CD4 binding, which represents a potential mechanism of virus neutralization. However, the V5 loop is highly variable in terms of length, amino-acid sequence, and the number and location of N-linked glycosylation sites[38–40]. Consequently, antibodies targeting this site are highly strain-specific. Introduction of a glycan at position 465 (T465N mutation) in BG505 Env-pseudotyped virus results in a strong reduction of serum neutralization titers ($ID_{50}$) for samples collected from both groups of animals at the week 38-time point (Supplementary Table 5). In the original BG505 sequence, this site is occupied by threonine and is directly interacting with the two abovementioned C3/V5-targeting pAbs. Consistently, in Rh.32034 and Rh.33311 samples the T465N mutation caused a decrease in NAb titers by ~35 and ~15-fold, respectively. Altogether, our findings further support that BG505 SOSIP-elicited antibodies targeting the C3/V5 epitope are the primary contributors to autologous neutralization in the rhesus macaque animal model.

**Antibodies targeting the N241/N289 glycan hole epitope.** The N241/N289 glycan hole in BG505 is an immunodominant epitope for autologous BG505-specific antibodies[20]. Although absent

from the BG505 sequence, the N241 and N289 potential N-linked glycosylation sites (PNGS) exist in 97% and 79% of known HIV-1 strains, respectively[20,25]. The Grp 1 immunogen, BG505 SOSIP MD39, did not include glycosylation sites at positions 241 and 289, and four out of six animals developed antibodies targeting this epitope (Fig. 1). N241 and N289 glycans were present in the Grp 2 immunogen, BG505 SOSIP.v5.2 N241/N289, which reduced antibody responses against this epitope. Interestingly, one animal in Grp 2 (Rh.33203) still developed antibodies targeting the N289 epitope. This is likely a consequence of incomplete glycosylation at the engineered N289 PNGS in BG505 SOSIP. Site-specific glycan analysis data reported here (Supplementary Table 6) and elsewhere[4,15] suggests that the N289 site can be up to ~40% unoccupied while the glycosylation efficacy at the N241 site is >90%.

CryoEMPEM analysis of polyclonal antibodies from Grp 1 animals Rh.32034 and Rh.33104 complexed with the BG505 SOSIP MD39 antigen yielded three high-resolution maps of pAbCs targeting the N241/N289 glycan hole epitope cluster (Fig. 3B, Supplementary Fig. 7B). In sample Rh.33104, two antibodies were computationally sorted, including an N241-directed antibody not previously detected using nsEMPEM. The reconstructed antibodies targeted partially overlapping epitopes but had distinct binding modes. Rh.33104 pAbC-2 was biased toward the N241 glycan hole (residues 83–86 in C1, and residues 229–232, 239–243, 267–269 in C2). Conversely, Rh.33104 pAbC-1, elicited in the same animal, is biased toward the N289 glycan hole (residues 266–270 and 289–290 in C2, as well as the C3/V4 interface and glycans N355 and N398). Rhesus macaque mAbs targeting similar sites on BG505 SOSIP.664 have been recently described and structurally characterized[29]. The third pAb, Rh.32034 pAbC-2, featured a different angle of approach compared to the other two and made contact with both, the N241 (residues: 83–85, 227–231, 240–243) and N289 (residues 289–290 and 267–269) regions of the glycan hole epitope.

Consistent with previous findings, our data suggest that the N241/N289 glycan hole epitope in BG505 is immunodominant due to poor glycan shielding. High-resolution analysis revealed that the elicited antibodies can utilize different angles of approach and different combinations of heavy and light chain CDRs to make contact with the exposed peptidic surface consisting of residues from C1, C2, C3, and V4 regions. Serum neutralization experiments performed with mutated BG505 pseudovirus showed that glycan knock-ins at either position N241 or N289 resulted in an only a minor reduction of ID$_{50}$ titers for Grp 1 samples (Supplementary Table 5). These results, as well as previous observations[20], suggest that glycan hole-directed antibodies elicited in rhesus macaques do not contribute significantly to autologous neutralization of BG505 virus. In contrast, antibodies targeting this site have been found to be major contributors to polyclonal rabbit serum neutralization[14,20,25].

**Antibodies targeting the fusion peptide and N611-glycan epitopes.** Antibodies against the fusion peptide and proximal N611-glycan epitope were detected in three out of six animals in Grp 2 at week 26 (Fig. 1D). In animal Rh.33311 both types of responses were elicited. Polyclonal complexes with a single pAbC targeting each site were computationally sorted from the cryoEMPEM data (Fig. 2B), and structural models were derived from each reconstructed map (Fig. 3C, D; Supplementary Fig. 7C, E).

Rh.33311 pAbC-1 utilizes all three HCDR loops to capture the N-terminal part of the fusion peptide (residues 512–520). Additional contacts are made to the fusion peptide proximal region (FPPR, residues 532–536) and the 664-helix in HR2 (residues 648–655). The light chain contributes only minimally to

antigen interaction. Glycans at positions N611, N241, and N88 are visible on the map, but they do not make substantial contact with the antibody.

The fusion peptide epitope is a well-explored target for bnAbs and is the focus of ongoing HIV vaccine design efforts[41–43]. We compared the Rh.33311 pAbC-1 structure to the available bnAb structures (Supplementary Fig. 8A). When looking at the angle of approach, this pAbC showed the most similarity to DFPH-a.15, a vaccine-elicited fusion peptide antibody isolated from Rhesus macaques[41]. However, the usage of heavy and light chains in binding is dissimilar between the two antibodies and the FP conformation in the Rh.33311 pAbC-1 bound state is more similar to ACS202, another FP-directed bnAb[44].

The N611-glycan epitope is proximal to the FP[29]. Importantly, this PNGS is often only partially glycosylated in BG505 SOSIP constructs[15,45] leading to the elicitation of narrow-specificity antibodies that preferentially recognize BG505 antigens lacking the N611 glycan. Some vaccine-elicited antibodies (e.g., RM20E1) are capable of neutralizing BG505 pseudotyped virus with N611 glycan knockout but not the wt BG505 pseudovirus[29]. CryoEM-PEM analysis of pAbs from animal Rh.33311 yielded one antibody binding to this site (Rh.33311 pAbC-2, Fig. 3D). Despite somewhat lower map resolution (~4.5 Å), the density corresponding to N611 glycan is clearly discernible in the binding site. In fact, the Rh.33311 pAbC-2 appears to be making extensive contact with the N611 glycan via the HCDR loops 1 and 3 and LCDR loop 2. The peptide part of the epitope consists of HR2 residues 611–618 as well as the N terminus of gp120 (residues 30–33).

Antibodies targeting the N611 or FP epitopes were observed in one-third of animals immunized with BG505 SOSIP.v5.2 N241/N289 trimers, but combined analysis of pseudovirus inhibition data suggest that they provide only minor (if any) contribution to autologous neutralization. Knock-out of the N611 glycan in the BG505 pseudovirus (N611A mutation) resulted in a subtle increase in serum neutralization titers for animal Rh.33311 compared to the wt BG505 pseudovirus (Supplementary Table 5). A similar trend is observed in the other two animals from Grp 2 with N611- and/or FP-directed antibodies (Rh.34686 and Rh.33182). Further studies with isolated mAbs are required to better characterize the two pAb families observed in animal Rh.33311.

**Antibodies targeting the gp120–gp120 interface epitope.** Gp120–gp120 interface antibodies were detected in all six animals from Grp 2 but in none of the animals from Grp 1. CryoEMPEM analysis of animal Rh.33311 yielded two structurally unique polyclonal classes of antibodies with partially overlapping footprints (Fig. 4A, B and Supplementary Fig. 7D). Rh.33311 pAbC-3 uses both the heavy and light chain to make contact with the C1 loop (residues 58–71) and the N262 glycan. Antibody binding induces the folding of this typically disordered region of the C1 loop into a short helix (Fig. 4A). On the other hand, Rh.33311 pAbC-4 primarily interacts with the V3 tip (residues 304–320) using the heavy chain (Fig. 4B). Binding to this epitope requires avoiding the N197 glycan, achieved by a slight rotation (~9°) of gp120 relative to the adjacent protomer, breaking the trimer symmetry (Supplementary Fig. 8B). Interestingly, the LCDR 3 loop of Rh.33311 pAbC-4 makes contact with the CD4 binding site residues 429 and 430; CD4bs is a highly pursued epitope target for HIV vaccine design[2,46–48].

Interestingly, both antibodies bind to epitopes that have not previously been described. Closer examination, however, reveals that they make significant contacts with non-native stabilizing residues introduced during the immunogen design process.

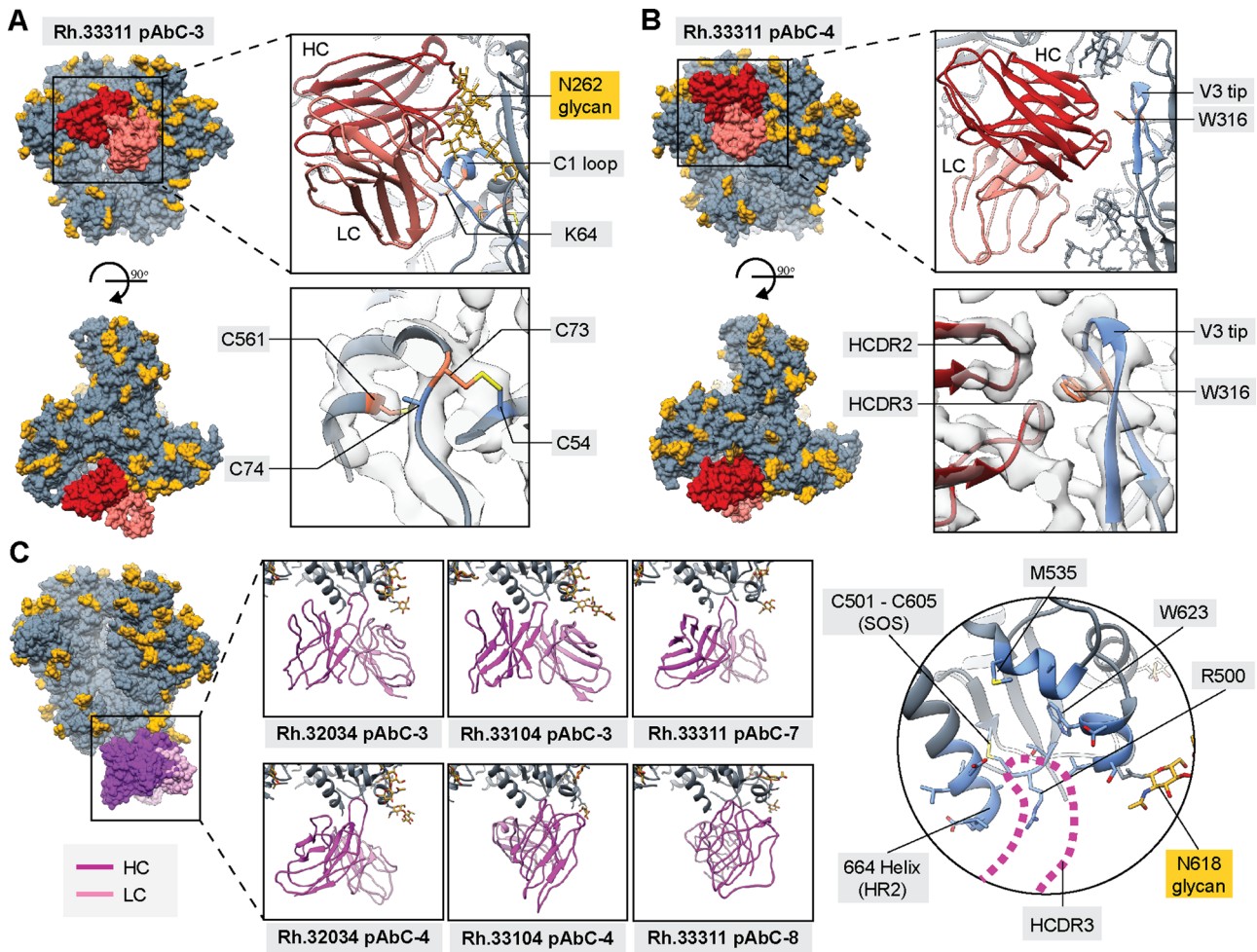

**Fig. 4 Structural analysis of non-neutralizing antibody responses.** Structures of BG505 SOSIP antigens in complex with polyclonal antibodies targeting gp120–gp120 interface (**A**, **B**) and base epitopes (**C**). BG505 SOSIP trimers are shown in gray with N-linked glycans in golden yellow. Antibodies are colored using the same scheme as in Figs. 1 and 2. Inferred heavy and light chains for each antibody are represented in different colors and labeled in each panel (HC and LC, respectively). The most relevant epitope/paratope components are indicated in the panels. Surface representation is used for the full immune complex (left side of each panel). Trimer residues in direct contact with the antibodies are colored blue (ribbon and stick representation). Engineered stabilizing residues are represented as sticks and colored orange. EM Map is displayed as transparent light gray surfaces in panels (**A**) and (**B**).

Rh.33311 pAbC-3 interacts with residue K64 (E in the original BG505 sequence; Fig. 4A, top right panel). In addition, in the immediate proximity to the binding site, engineered cysteine residues C73 and C561 appear to be cross-linked with the native disulfide bridge usually formed between residues C54 and C74 (Fig. 4A, bottom right panel). In the high-resolution cryoEM maps of BG505 SOSIP.v5.2 N241/N289 constructs reported here, we have found evidence of an altered disulfide bond network at this site (i.e., the pairing of C73 with C54 and C74 with C561). This change induces significant conformational rearrangements in the C1 loop on the level of both, side chain and main chain (Supplementary Fig. 8C). The most dramatic change is observed within the HR1 region surrounding residue C561 (amino acids: 560–567), which gets restructured into a short helix. However, the formation of the disulfide bond network at this site is stochastic in nature. This is supported by the low local resolution for this region across all Rh.33311 cryoEM maps, consistent with a high degree of conformational heterogeneity. Furthermore, we have previously reconstructed EM models of BG505 SOSIP.v5.2 with the "intended" pairing of cysteines[49]. The altered disulfide network together with the E64K mutation, therefore, comprises a neo-epitope not present on wt BG505 Env. Similarly, the Rh.33311 pAbC-4 binds the V3 tip, with the HCDR 2 and 3

contacting residue W316 (Fig. 4B, top and bottom-right panels). Tryptophan was introduced in place of alanine at this position beginning in SOSIP.v4 constructs to stabilize the trimers in a closed, pre-fusion state[10,50].

We next tested whether the interface-directed antibodies could bind to BG505 in the absence of these stabilizing mutations. Using BG505 SOSIP.v3 trimers that were not engineered with the abovementioned C1 and V3 mutations, we conducted nsEMPEM on the Rh.33311 polyclonal sample; experiments with BG505 SOSIP.v5.2 N241/N289 (the immunogen) was also performed in parallel for comparison (Supplementary Fig. 2C). We observed antibodies to C3/V5, N611, FP, and base epitopes with either antigenic probe, suggesting that these epitopes are not significantly affected by the engineered residues. However, interface-specific antibodies were not detected with BG505 SOSIP.v3, indicating that their binding is dependent on the presence of stabilizing mutations. Consequently, these classes of antibodies are unable to recognize native Env trimers or neutralize the virus.

We overlaid the two pAbC models with structures of SOSIP complexes featuring VRC01 (PDB ID: 6V8X[51]) and SF12 mAbs (PDB ID: 6OKP[52]), two bnAbs targeting the CD4bs and the silent face epitopes, respectively (Supplementary Fig. 8D). Both, Rh.33311 pAbC-3 and pAbC-4 have partially overlapping epitope

footprints and clash with the VRC01 and SF12 bnAbs. These data show that the gp120–gp120 interface antibodies could potentially interfere with the elicitation of antibody response against the CD4bs and the silent face epitopes.

**Antibodies targeting the trimer base epitope.** The trimer base is the most immunodominant epitope cluster on BG505 SOSIP trimers[29–31]. In native membrane-bound Env trimers, this part of the molecule is connected to the transmembrane domain and shielded from antibodies by the viral membrane[28]. From the three cryoEMPEM datasets (Rh.32034, Rh.33104, and Rh.33311), we recovered six high-resolution maps with base-targeting pAbCs (Fig. 2, Supplementary Fig. 7F). Analysis of the reconstructed EM maps and modeled structures revealed that antibodies can approach the base at many different angles (Fig. 4C, left). Surprisingly, all six pAbCs converge on a common epitope and share similar binding mechanisms. Base-specific pAbCs utilize HCDR3 to make contact with the binding pocket located at the C terminus of BG505 SOSIP (664-helix in HR2; Fig. 4C, right). These findings are consistent with previous studies of the base-targeting mAbs RM20A3 and RM19R, isolated from rhesus macaques immunized with BG505 SOSIP.664[29,53].

**Design and characterization of two-component nanoparticle system for presentation of BG505 SOSIP immunogens.** To explore the effects of nanoparticle presentation of BG505 SOSIP trimers on the epitope specificities of vaccine-elicited antibodies, we used the computationally designed tetrahedral nanoparticle scaffold T33-31[54,55]. This nanoparticle consists of four copies each of two complementary trimeric components, referred to as T33-31A and T33-31B (Supplementary Fig. 9A). Both components have outward-facing N termini that were genetically fused to the C terminus of BG505 SOSIP.v5.2(7S) N241/N289, using a flexible GS linker (Fig. 5A, Supplementary Fig. 9A). nsEM analysis of the BG505 SOSIP-T33-31A and -T33-31B antigen-bearing components confirmed that genetic fusion did not result in major issues with antigen folding and stability (Supplementary Fig. 9B). Additional density corresponding to the nanoparticle component is clearly discernible in 3D reconstructions of the two antigen-bearing components.

The BG505 SOSIP-T33-31 nanoparticle was assembled by combining equimolar amounts of BG505 SOSIP-T33-31A and BG505 SOSIP-T33-31B. Analysis using cryoEM showed that nanoparticles assembled as expected with eight flexibly linked trimer antigens on their surface (Fig. 5B, top row). We processed the molecular projection data by independently analyzing signal originating from three flexibly bound objects: T33-31 nanoparticle core, BG505 SOSIP fused to component A and BG505 SOSIP fused to component B (Fig. 5B, middle and bottom rows). The T33-31 nanoparticle core structure closely matched the published crystal structure (PDB ID: 4zk7[55]); with a backbone RMSD of 0.65 Å, confirming that antigen attachment did not affect nanoparticle assembly. Additionally, the analysis of the two BG505 SOSIP structures (A and B) revealed that the presented trimer antigens are folded correctly in the native-like, pre-fusion state.

Consistent with the structural data, biolayer interferometry (BLI) experiments demonstrated that the antibody binding profiles of BG505 SOSIP fused to each nanoparticle were equivalent to those of free trimers (Supplementary Fig. 9C). However, reduced binding to antibodies targeting epitopes in the middle/lower part of the trimer such as the fusion peptide (PGT151, VRC34, and ACS202), gp120–gp41 interface (35O22), gp41–gp41 interface (3BC315), and the trimer base (RM19R and RM20A3) was observed in the context of the assembled

nanoparticles. This is likely caused by the crowding of trimer antigens on the nanoparticle surface, limiting the accessibility of epitopes near or at the trimer base; an observation consistent with previously reported data using other nanoparticle scaffolds[14–16].

**Immunization experiments with BG505 SOSIP-T33-31 nanoparticle.** After confirming the presence of appropriate antigenic and structural features in the designed BG505 SOSIP-T33-31 nanoparticle, we proceeded to use this reagent for rhesus macaque immunizations. We tested whether presentation of BG505 SOSIP on the T33-31 nanoparticles could have a detectable effect on antibody-mediated immune response compared to immunization experiments with soluble BG505 SOSIP trimers (Fig. 6A; Grp 3 of animals). The immunization experiments were performed at the same time and under the same experimental conditions as with Grps 1 and 2. We used ELISA and TZM-bl pseudovirus inhibition assays to determine the relative proportions of vaccine-elicited antigen-specific and NAbs, respectively (Fig. 6B, Supplementary Tables 1 and 2). ELISA midpoint titers were comparable to those obtained for Grps 1 and 2 at all time points, suggesting that antibody responses of similar magnitude were elicited against the BG505 SOSIP antigen. Serum neutralization titers were somewhat lower compared to Grps 1 and 2, with only two animals developing detectable neutralization titers ($ID_{50} > 40$) after all four immunizations.

Next, we applied low-resolution polyclonal epitope mapping experiments to assess if nanoparticle presentation altered the epitope specificity/distribution of vaccine-elicited pAbs. The nsEMPEM analysis was performed on polyclonal samples from all Grp 3 animals at weeks 8, 10, 26, and 38 (Fig. 6C, Supplementary Fig. 10). Similar to Grps 1 and 2, all animals developed antibodies targeting the base of the BG505 SOSIP trimer. In five of six animals, these antibodies were observable after the first immunogen dose. C3/V5-targeting antibodies were detected in animal Rh.33065 at weeks 26 and 38. Serum neutralization was also detected for this animal at those two time points. In contrast to the soluble trimer immunization experiments, knock-in of the N465 glycan site to BG505 Env did not significantly impact NAb titers for this animal (Supplementary Table 5), indicating that antibody responses to other (non-C3/V5) epitopes also contribute to neutralization. Finally, four of six animals developed antibodies to the FP and N611 epitopes after all four immunogen doses. This is similar to the nsEMPEM results from Grp 2 but different from Grp 1.

The greatest difference for Grp 3, compared to Grps 1 and 2, was observed with polyclonal antibodies directed to the V1/V2/V3 epitope cluster (Fig. 6C). These antibodies were elicited by all animals after the first immunization, and they appear to target a range of epitopes comprising the V1–V3 variable loops at the trimer apex. Despite some similarity to apex-specific bnAbs (e.g., PGT121 and PG9), the elicited antibodies appear to lack strong neutralizing activity (Supplementary Table 2). However, animal Rh.33172 developed NAb titers ($ID_{50} = 407$) after four immunizations and nsEMPEM analysis revealed the presence of pAbs against two epitope clusters across all time points, namely the V1/V2/V3 and the base, suggesting that at least some of the V1/V2/V3-directed antibodies were neutralizing.

**CryoEMPEM analysis of pAb responses in animal Rh.33172.** Polyclonal Fabs from animal Rh.33172 (week 38) were complexed with BG505 SOSIP.v5.2(7S) N241/N289 (matched to the immunogen presented on T33-31 nanoparticle) and subjected to cryoEMPEM analysis. The analysis yielded five maps, each featuring a unique pAb class (Fig. 7A). Four pAbC targeted partially overlapping but distinct V1/V2/V3 epitopes and one antibody

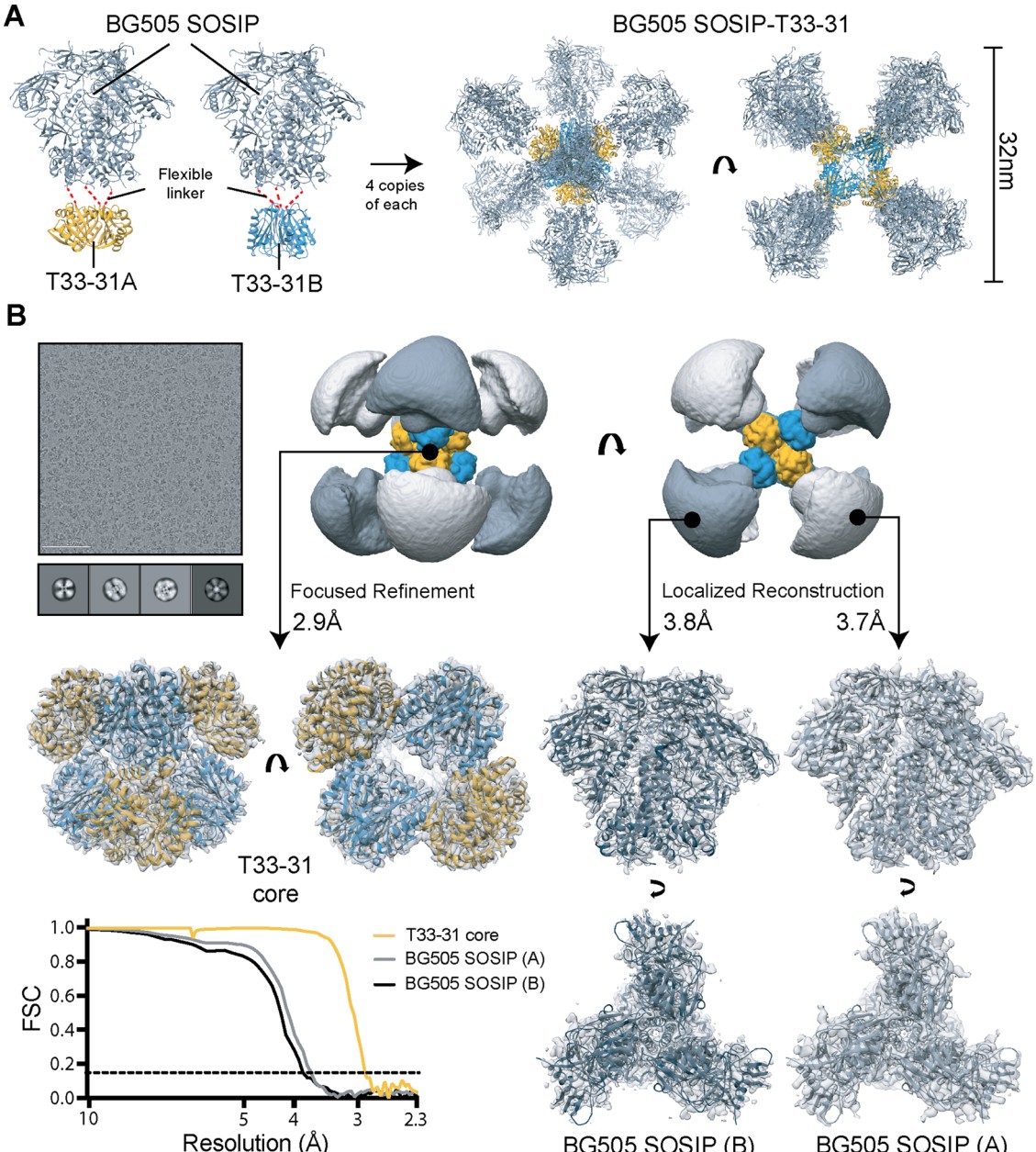

**Fig. 5 Design and characterization of the nanoparticle immunogen. A** Models of the two antigen-bearing components (left) and BG505 SOSIP-T33-31 nanoparticle (right). **B** CryoEM analysis of the designed BG505 SOSIP-T33-31 nanoparticle. Representative raw micrograph and sample 2D class-averages of imaged particles are shown in the top left corner. The scale bar in the micrograph is 100 nm. Low-pass filtered map of the entire nanoparticle is shown in the top right corner (orange—T33-31A; blue—T33-31B; light gray—BG505 SOSIP (**A**); dark gray—BG505 SOSIP (**B**)). Final maps and relaxed models from subparticle analysis are shown in the middle/bottom panels. Fourier shell correlation (FSC) curves for each subparticle map are presented in the bottom left corner. The cryoEM experiments have been performed once.

class was directed against the trimer base. The average apparent resolution of the reconstructed EM maps was ~4.0 Å (Fig. 7A, Supplementary Fig. 11). Analysis of the maps and corresponding models revealed that the Rh.33172 pAbC-2 epitope consists mainly of V2 and V3 residues, while the other three antibodies primarily interact with the V1 loop, making minor additional contacts with V2 and V3 (Fig. 7B).

Rh.33172 pAbC-1 and pAbC-3 approach the V1 loop from different angles, but both converge on the highly conserved N-linked glycosylation site N156[38–40]. Strikingly, close inspection of the data revealed no evidence of glycan density at this site in both maps (Fig. 7B, left panels). Furthermore, for Rh.33172 pAbC-1, no glycan density was found at N137, a residue that is also within

the binding site for this antibody. These results suggest that both classes of antibodies target a small subset of BG505 SOSIP antigens that lack N-linked glycans at one or both of the PNG sites. These pAbs share high similarity to rhesus macaque mAbs elicited with the BG505 SOSIP-based immunogen RC1, in which the conserved V1 glycans: N133, N137, and N156, were knocked out to improve accessibility at this epitope[9]. Site-specific glycan analysis data for the BG505 SOSIP.v5.2(7S) N241/N289 construct (Supplementary Table 6) reveal that while N137 can be up to 50% unoccupied, the N156 PNG site is glycosylated in >89% of the protomers. Furthermore, the N156 site appears to be fully glycosylated in the BG505 SOSIP-T33-31A and -T33-31B components that make up the nanoparticle immunogen. Given

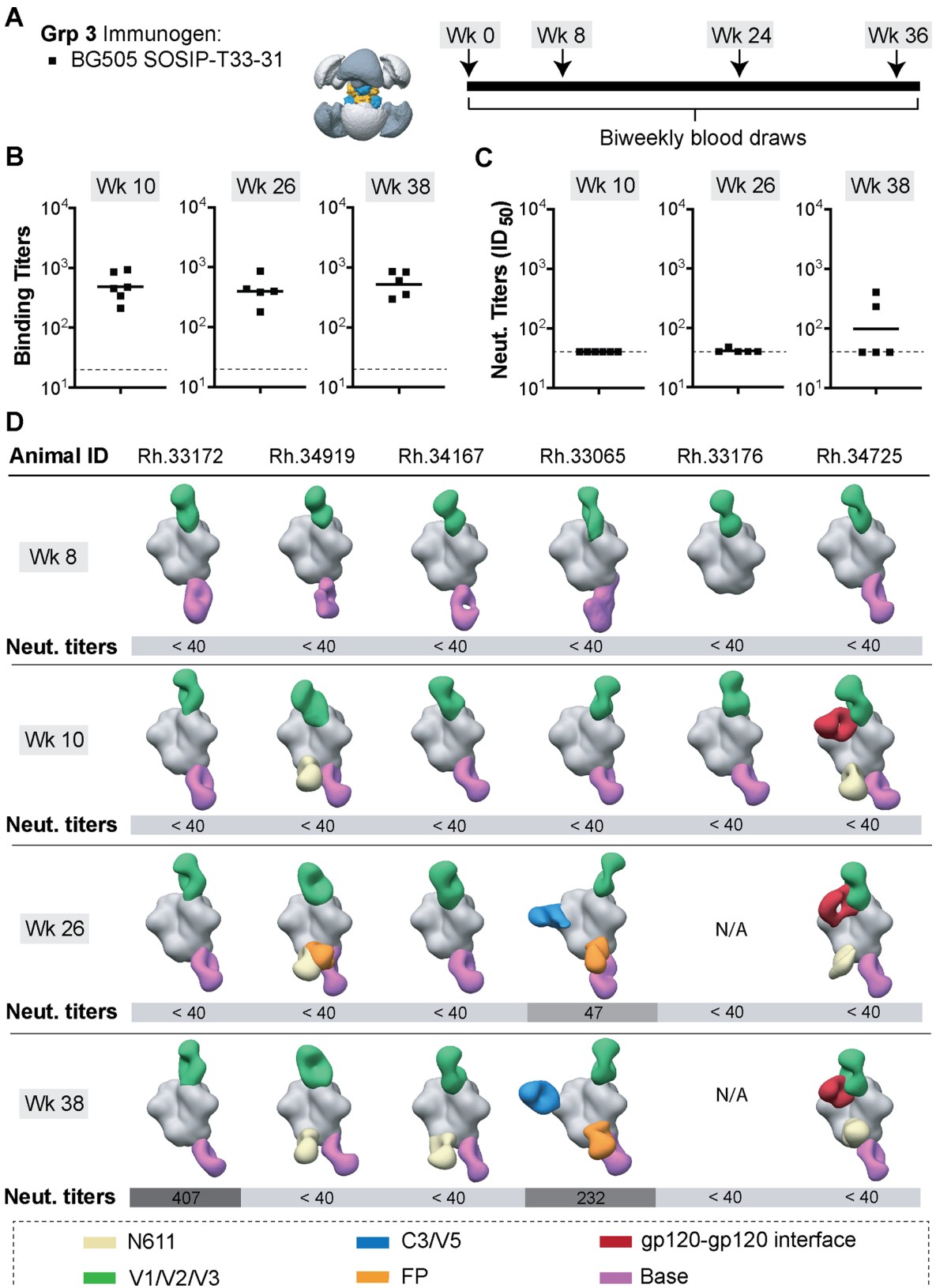

**Fig. 6 Immunization experiments with the octameric BG505 SOSIP-T33-31 nanoparticle. A** Immunization information and schedule. **B** ELISA binding titers (midpoint) and **C** neutralizing antibody titers (ID$_{50}$) for plasma samples collected at time points indicated above each graph. Horizontal lines represent the geometric mean values at each time point ($n = 6$ animals for week 10 and $n = 5$ animals for weeks 26 and 38; see Method section for details). ELISA and pseudovirus neutralization experiments were performed in triplicates ($n = 3$) for each serum sample at every time point and the mean value is plotted. **C** Composite figures from nsEMPEM analysis of polyclonal responses at weeks 8, 10, 26, and 38. Animal IDs are shown above each dataset. Neutralizing antibody titers for corresponding animals are shown below each composite figure. A color-coding scheme for antibodies targeting different epitope clusters is shown at the bottom. BG505 SOSIP antigen is represented in gray.

the relatively small amount of glycan underoccupancy, it is surprising that antibodies are readily induced against this site.

We tested if knocking out the N156 glycosylation site on a BG505 Env would make the pseudovirus more susceptible to neutralization by sera from this group of animals. Four out of five animals, including Rh.33172, had increased serum neutralization titers ($ID_{50}$) against the N156 mutant pseudovirus compared to the wt BG505 (Supplementary Table 5). This finding supports our observations from the structural data.

Analysis of the Rh.33172 pAbC-2 structure revealed that this pAb class utilizes a long, 22-amino-acid HCDR3 to interact with the V3-tip (residues 304–321). The antibody coordinates glycans N301 and N156 in BG505 and contacts the V2 loop (residues 170–173). This epitope is similar to the Rh.33311 interface-targeting pAbCs; it is partially constituted of engineered residues. In the BG505 SOSIP.v5.2(7S) N241/N289 construct there are three stabilizing mutations within the V3-loop (R304V, A316W, and A319Y), and Rh.33172 pAbC-2 makes contact with all of the mutated residues (Fig. 7B, right). nsEMPEM analysis revealed that Rh.33172 pAbC-2 failed to bind to BG505 SOSIP.v3, which lacks the stabilizing mutations in V3 (Supplementary Fig. 2D, Rh.33172 pAbC-2 depicted in olive green). Consequently, this class of polyclonal antibodies is non-neutralizing because they are unable to recognize wt BG505 Env.

Finally, the Rh.33172 pAbC-4 makes extensive contacts with the V1 loop (residues 132–156) as well as the V2 loop (residues 186–190) while also coordinating the V1 glycans (N133, N137, and N156). This antibody class can interact with the BG505 SOSIP.v3 antigen (Supplementary Fig. 2D; depicted in light green) and is likely responsible for the serum neutralization observed for animal Rh.33172 at the week 38 time point. Previous studies have shown that V1-targeting antibodies in BG505 can be moderately neutralizing[20,28,35].

## Discussion

In this study we describe cryoEMPEM, a method for high-resolution mapping of vaccine-induced pAb responses directly from sera. Isolation and analysis of mAbs is the gold standard for structural characterization of antibodies and antibody–antigen complexes. However, cryoEMPEM is a faster alternative offering a more comprehensive view of diverse pAbC targeting an antigen at the serum level. On average, it takes 2–3 days to isolate polyclonal Fabs from serum and ~2 days to assemble immune complexes and prepare cryoEM grids. Multiple samples can be processed in parallel during these early steps. The imaging of vitrified immune complexes and subsequent data processing are dependent on the specific equipment configuration in different institutes/labs and can take anywhere from one week to one month. Under ideal conditions (i.e., access to state-of-the-art imaging and computational resources), cryoEMPEM analysis can be completed within ~10 days from serum/plasma collection. Therefore, it can be applied to study immunization progress in real-time and make appropriate regimen changes, when needed.

Compared to structural characterization of mAbs, there are additional experimental challenges caused by the polyclonal nature of bound antibodies: compositional heterogeneity in the EM data and the lack of sequence information. Using the described focused classification approach, we reconstructed 21 maps of different pAbC with apparent resolution averaging ~4.1 Å, from the immune sera of four animals. The quality and resolution are comparable to cryoEM maps of immune complexes with mAbs. In addition, high-resolution analysis allowed us to resolve multiple structurally unique pAbC that interact with overlapping epitopes within the same polyclonal sample. The maps were of sufficient quality to relax pseudo-atomic models

and while the explicit antibody sequences are inherently unknown, the models represented as poly-alanine peptide backbone provided insight into different aspects of antibody binding (i.e., all epitope contacts on the antigen side, and usage of heavy and light chain and different CDRs on the antibody side).

Our study included the comparison of three different BG505-based immunogens and revealed critical differences in the immune response elicited by each design. The three immunogens primarily differed in their content of stabilizing mutations and glycosylation sites. BG505 SOSIP MD39 trimers elicited N241/N289 glycan hole responses in the majority of animals in Grp 1. This highly specific and undesirable response is almost fully absent in animals receiving the other two immunogens (Grps 2 and 3) because of the engineered PNG sites at positions N241 and N289. We speculate that glycan masking of the immunodominant glycan hole epitope in these immunogens drove the development of antibodies against more diverse sites (e.g., FP, N611, and V1/V2/V3 sites). Similar observations have been reported previously with suppression of immune response to the trimer base epitope and epitopes created by trimer degradation[7,18]. However, it is still unclear if immuno-masking of all immunodominant autologous epitopes in BG505 SOSIP would lead to elicitation of antibody responses targeting poorly accessible, cross-reactive sites.

The SOSIP.v5.2 stabilizing mutations in V3 (A316W) and C1 (E64K, A73C) in the Grp 2 and 3 immunogens generated neo-epitopes that were immunogenic. The stabilization of protein domains has been found to reduce the entropic cost of binding to other proteins (including antibodies) and ultimately enhance affinity[56–58]. We have previously shown that rigidification of HCDR3 loops in the precursors of HIV bnAbs can drive affinity maturation[59]. In addition, bioinformatic studies of antibody-antigen interaction networks have revealed that aromatic amino acids (e.g., Trp, Tyr, and Phe) are enriched in both, epitopes and paratopes[60–63]. They contribute to binding by forming energetically favorable hydrophobic and van der Waals interactions. We believe that these two mechanisms (stabilization of flexible protein domains and surface-exposure of non-native aromatic amino acids) resulted in efficient elicitation of antibody response against the V3 and C1 loops. Nevertheless, we demonstrated that antibodies targeting these sites are unable to interact with antigen in the absence of stabilizing mutations and consequently with the wt BG505 Env on a virion. They are therefore non-neutralizing. Besides the potential immunodominance of these neo-epitopes, they are also problematic because the antibodies targeting them could potentially sterically block access to bnAb epitopes (e.g., CD4bs and V2-apex).

While nanoparticle display did not improve the overall levels of NAb responses, it had a dramatic effect on the epitopes targeted and angles of approach of elicited antibodies, particularly after early immunizations. This finding is consistent with previous studies performed with ConM and BG505-based nanoparticle immunogens[14,15,64]. While immuno-focusing on the trimer apex is a desirable feature, some of the antibody responses we observed targeted non-native glycan holes that were present on the recombinantly expressed antigen, even at very low levels. This result further emphasizes the need to conduct glycopeptide occupancy analysis of Env immunogens. Interestingly, our analyses revealed that presentation on the nanoparticle failed to eliminate base-directed antibody responses, suggesting that this epitope is still readily accessible on the nanoparticle or becomes accessible upon nanoparticle disassembly in vivo.

This work provides a comprehensive reference for HIV vaccine design efforts utilizing BG505 SOSIP immunogens. The presented data can be readily applied to engineer changes that would immuno-silence undesirable epitopes and/or immuno-focus to a specific site in this construct. More broadly, cryoEMPEM is a

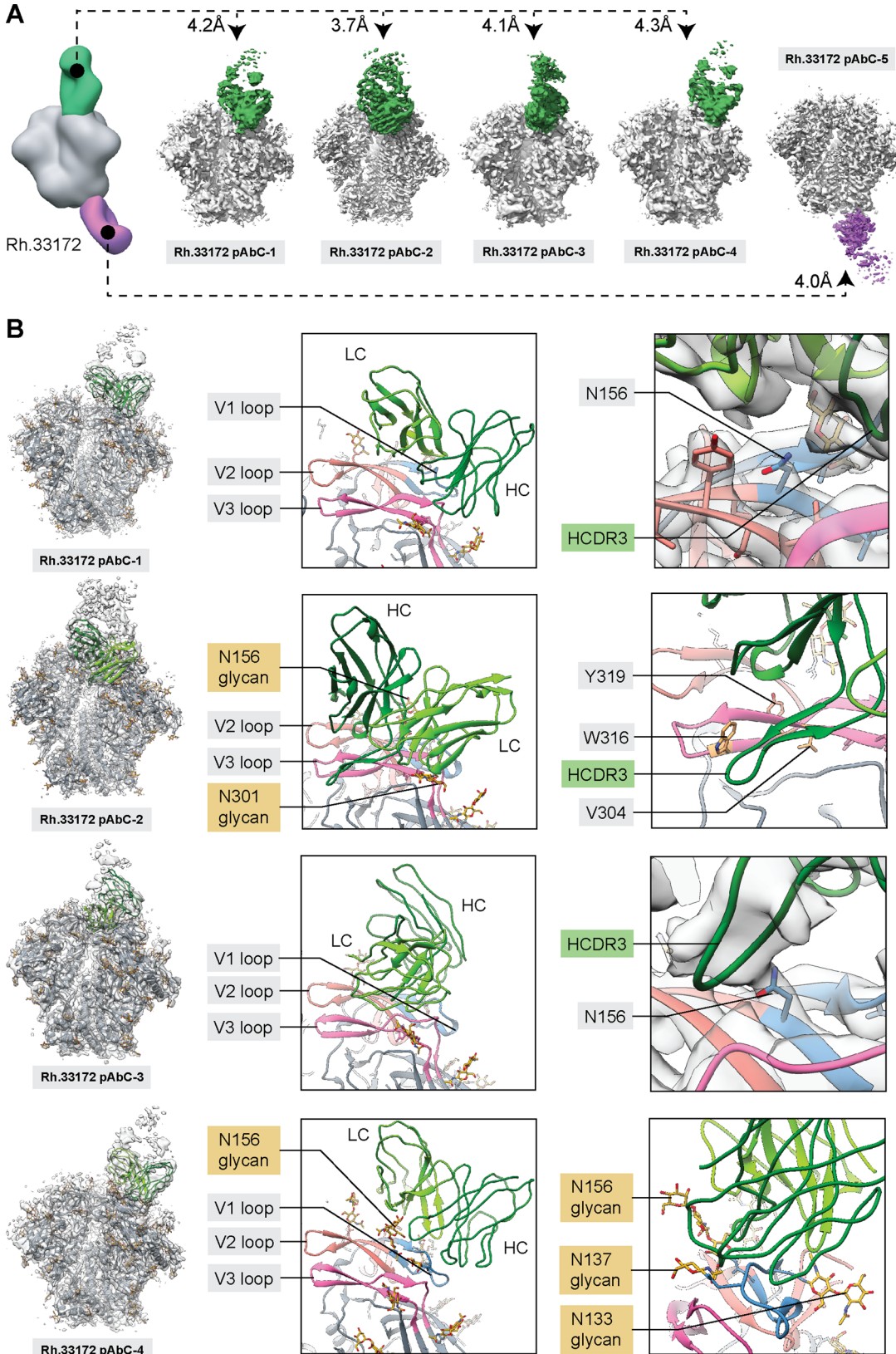

**Fig. 7 CryoEMPEM analysis of polyclonal antibodies elicited by BG505 SOSIP-T33-31 nanoparticle immunogen. A** CryoEMPEM analysis of the immune complexes generated using polyclonal Fabs isolated from animal Rh.33172 (week 38 plasma samples). BG505 SOSIP antigen is represented in gray and Fab densities are colored according to the scheme used in Fig. 6. The apparent resolution of each reconstructed map is indicated. **B** Structures of four V1/V2/ V3-targeting pAbs. Models (ribbon representation) and maps (light gray mesh) are shown on the left, and close-up views of each structure with the most relevant epitope/paratope components are indicated (V1 loop—blue, V2 loop—salmon, V3 loop—magenta). Inferred heavy and light chains for each antibody are labeled in each panel and represented in a dark and light green color, respectively.

general method that can be applied to different subunit vaccine platforms for rapid and detailed mapping of antigenic landscapes at high resolution. With continued methodological advances, it may be possible to directly determine the sequences of bound polyclonal antibodies by cryoEMPEM.

## Methods

**DNA vectors and cloning.** Constructs containing BG505 SOSIP.v5.2 N241/N289 and BG505 SOSIP.v5.2(7S) N241/N289 genes codon-optimized for mammalian cell expression were subcloned into a pPPI4 vector. BamHI and NheI restriction sites were used for insertion of genes encoding T33-31A and T33-31B nanoparticle assembly components to the C-terminus (residue 664) of BG505 SOSIP.v5.2(7S) N241/N289. Ten-amino-acid flexible linkers (sequence: GSGSGSGSGG) were introduced between the SOSIP trimer and each nanoparticle component. Restriction enzymes and Quick Ligation kits were obtained from New England Biolabs (NEB). Sequences were verified by Sanger sequencing (Genewiz). Nucleotide sequences of the constructs used in this study are presented in Supplementary Table 7.

**Expression and purification of BG505-SOSIP and antigen-bearing components.** BG505 SOSIP-based constructs were purified as described previously[15,19]. Briefly, BG505 SOSIP.v5.2 N241/N289, BG505 SOSIP.v5.2(7S) N241/N289, BG505 SOSIP-T33-31A, and BG505 SOSIP-T33-31B constructs were expressed in HEK293F cells (Invitrogen) and purified using PGT145 immunoaffinity chromatography. BG505 SOSIP MD39 was expressed in FreeStyle 293F cells (Thermo Fisher Scientific) and purified using a 2G12 immunoaffinity column. Samples were eluted from the corresponding immunoaffinity matrix with 3 M $MgCl_2$ buffer and subjected to size-exclusion chromatography (SEC) on a HiLoad® 16/600 Superdex® pg200 (GE Healthcare). Samples were subsequently concentrated to 1 mg/ml and stored frozen in DPBS (Thermo Fisher Scientific).

**BG505 SOSIP-T33-31 nanoparticle assembly and purification.** The two antigen-bearing nanoparticle components (BG505 SOSIP-T33-31A and BG505 SOSIP-T33-31B) were independently purified using the protocol described above. Approximately, 2.5 mg of each component were combined and incubated for 72 h at 37 °C, to drive co-assembly into nanoparticles. Average nanoparticle yield after 72 h was ~70% of the starting material. Assembled nanoparticles were purified from residual unassembled components using Sephacryl S-500 HR column with DPBS (Thermo Fisher Scientific) as the running buffer. ToxinSensor™ Single Test Kit (GenScript) was applied to verify that the endotoxin levels of the nanoparticle sample were below 50 EU/kg per dose.

**Biolayer interferometry.** The experiments were performed as described previously[15,65]. Kinetics buffer (DPBS + 0.1% bovine serum albumin (BSA) + 0.02% Tween-20) was used to prepare antibody and antigen dilutions. IgG versions of antibodies were diluted to 5 µg/ml. The concentrations of trimer and nanoparticle samples were adjusted to equimolar concentration (500 nM) of the BG505 SOSIP.v5.2(7S) N241/N289 antigen in each sample. BG505 SOSIP.v5.2(7S) N241/N289 trimer was included as a reference. Octet Red96 instrument (ForteBio) was used for data collection. Antibodies were immobilized onto anti-human IgG Fc capture (AHC) biosensors (ForteBio). The lengths of association and dissociation steps were adjusted to 180 and 300 s, respectively. Octet System Data Analysis v9.0 (FortéBio) software package was used for data processing. The background was corrected by subtracting the negative control datasets (kinetics buffer). The baseline step immediately preceding the association step was used for the alignment of y-axes. An interstep correction between the association and dissociation steps was also introduced. Final data plots were prepared in Excel. Experiments were performed in duplicate to test reproducibility, but only one measurement is presented.

**Site-specific glycan analysis using mass spectrometry.** The experiments were performed with BG505 SOSIP-T33-31A and BG505 SOSIP-T33-31B antigen-bearing components as described previously[15,66]. A standard library for BG505 SOSIP.664 expressed in HEK293F cells was used to search the MS data. The relative amount of glycan at each site was determined by comparing the extracted chromatographic areas for different glycopeptides with an identical peptide sequence. For analysis, we applied precursor mass tolerance of 4 and 10 ppm for fragments with a false discovery rate of 1%. The relative amount of glycan at each site and the unoccupied proportion were determined by comparing the extracted ion chromatographic areas for different glycopeptides with an identical peptide sequence. Glycan analysis data for the underlying BG505 SOSIP.v5.2(7S) N241/N289 Env construct (as free trimer) used for the generation of antigen-bearing components has been published elsewhere[15]. It is presented in this study as a reference for comparisons with BG505 SOSIP-T33-31A and BG505 SOSIP-T33-31B.

**Immunization experiments.** Rhesus macaque immunizations and blood draws were performed at the Yerkes National Primate Research Center, Atlanta, GA, USA. All procedures were approved by Emory University Institutional Animal Care and Use Committee protocol 201700723, and animal care facilities are accredited by the U.S. Department of Agriculture and the Association for Assessment and Accreditation of Laboratory Animal Care International. Three groups of Indian Rhesus macaques (6 animals per group) were immunized at weeks 0, 8, 24 and 36 with BG505 SOSIP MD39 (Grp 1, 100 µg per dose), BG505 SOSIP.v5.2 N241/N289 (Grp 2, 100 µg per dose) and BG505 SOSIP-T33-31 nanoparticle (Grp 3, 119 µg per dose). Total dose was normalized to administer an equivalent molar concentration of BG505 SOSIP antigen across all groups. Immunogens were formulated with Matrix-M™ (Novavax, Inc.; 75 µg per dose) for the first three immunizations (weeks 0, 8, and 24). For the week 36 immunization, the adjuvant was changed to SMNP (Darrell Irvine lab, MIT). Totally, 750 U of SMNP was used per immunogen dose. Animals were immunized subcutaneously with antigen-adjuvant formulation. The dose was split in half and injected into both hind limbs. Blood draws were performed biweekly. Animal Rh.33176 was euthanized after the second round of immunizations (week 10 time point) due to the development of gastric cancer. Consequently, sera samples from later time points were not available for this animal.

**ELISA binding assays.** Sandwich ELISA experiments were performed using sera samples from weeks 6, 10, 26, and 38. All buffer additions and wash steps were performed using a BioStack Microplate Stacker system (BioTek). 12N antibody (as IgG) was diluted to 3 µg/ml and immobilized onto high-binding, 96-well microplates (Greiner Bio-One) for 2 h at room temperature. The plates were washed three times with TBS + 0.1% Tween-20 (TBST). Plates were blocked with TBS + 5% BSA + 0.05% Tween-20 overnight at 4 °C. Plates were washed three times with TBST before the addition of antigen solution (PBS + 1% BSA + 3 µg/ml of BG505 SOSIP). BG505 SOSIP constructs were matched to the antigen used for the immunization of each group of animals. Plates were treated with antigen for 2 h at room temperature and subsequently washed three times with TBST. Serial two-fold sera dilutions (starting at 1:20) were prepared in PBS and loaded onto the antigen-coated plates (2-h incubation at room temperature). Following three washes with TBST, AP-conjugated AffiniPure goat anti-human IgG, (Jackson Immunoresearch, Cat # 109-055-097) in TBS + 1% BSA was added for 1 h at room temperature. The detection antibody was diluted 1:4000. Plates were washed three times with TBST, followed by the addition of 1-Step PNPP Substrate Solution (Thermo-Fisher Scientific). Colorimetric endpoint development was allowed to proceed for ~1 h before termination by 2 M NaOH. Data were recorded on a Synergy H1 plate reader (BioTek) by measuring the absorbance at a wavelength of 405 nm. Midpoint titers were determined using Graphpad Prism software. Experiments were performed in triplicate.

**Pseudovirus inhibition assays.** Pseudovirus inhibition (neutralization) assays were performed with Env-pseudotyped viruses and TZM-bl cells. The experiments with wt BG505 Env presented in Fig. 1, Fig. 6, and Supplementary Table 2 were performed at The Scripps Research Institute, La Jolla, CA, USA, as described previously[22]. The experiments involving mutant BG505 Env, presented in Supplementary Table 5 were performed at Duke University Medical Center in Durham, NC, USA, as previously reported[67]. Serial threefold dilutions of sera samples were pre-mixed with Env-pseudotyped virus and added to TZM-bl cells. Starting sera dilutions were 1:40 for the first and 1:20 for the latter group of experiments. Midpoint neutralization titers ($ID_{50}$) were determined as the serum dilution at which pseudovirus infectivity was inhibited by 50%.

**nsEMPEM—preparation of Fab and complex samples.** Experiments were implemented as described previously[15,27]. Plasma samples (weeks 8, 10, 26, and 38) from three groups of six immunized rhesus macaques were chosen for polyclonal epitope mapping. IgGs were purified from ~1 ml of plasma with an equal volume of settled Protein A Sepharose resin (GE Healthcare). Samples were eluted from the resin with 0.1 M glycine at pH 2.5 and immediately neutralized with 1 M Tris-HCl pH 8. Amicon ultrafiltration units with a 30 kDa cutoff (Millipore Sigma) were used to concentrate and buffer exchange the purified IgG to the digestion buffer (PBS + 10 mM EDTA + 20 mM cysteine, pH 7.4). IgG samples were digested for 10 h at 37 °C using 50 µl of settled papain-agarose resin (Thermo Fisher Scientific). Fc and non-digested IgG were removed through a 1-hour incubation at room temperature with Protein A Sepharose resin using 0.2 ml packed resin per 1 mg of IgG. Fab samples were concentrated to ~6–8 mg/ml and buffer exchanged to TBS using Amicon ultrafiltration units with a 10 kDa cutoff (EMD Millipore Sigma). Final Fab yields were ~1 mg. The complexes were assembled using 1 mg of purified polyclonal Fab and 15 µg of the corresponding BG505 SOSIP antigen used in the immunization (BG505 SOSIP MD39 for Grp 1 samples; BG505 SOSIP.v5.2 N241/N289 for Grp 2 samples; BG505 SOSIPv5.2(7S) N241/N289 for Grp 3 samples). For Fab samples from Rh.33311 and Rh.33172 (week 26 time point), the complexing was also performed with BG505 SOSIP.v3 under equivalent conditions. The reactions were incubated for ~18 h at room temperature. Immune complexes were purified from residual Fab using SEC (Superose 6 Increase column) with TBS as a running buffer, concentrated with Amicon ultrafiltration units with a 10 kDa cutof, and immediately placed onto carbon-coated 400-mesh Cu grids as described in the nsEM section.

**Negative stain electron microscopy (nsEM)**. Negative stain electron microscopy experiments were performed as described previously[15,27]. Purified EMPEM complexes were diluted to 30–50 μg/ml and applied to carbon-coated 400-mesh Cu grids (glow-discharged at 15 mA for 25 s) for 10 s and then blotted. BG505 SOSIP-T33-31A and BG505 SOSIP-T33-31B trimer samples were diluted to 20 μg/ml and loaded onto grids following the same protocol. Grids were negatively stained using uranyl-formate, 2% (w/v), for 40 s. Data were collected on either a Tecnai Spirit electron microscope, operating at 120 keV, or a Tecnai TF20 electron microscope, operating at 200 keV. Nominal magnification was ×52,000, with a pixel size of 2.05 Å (at the specimen plane) for the Spirit and ×62,000, with a pixel size of 1.77 Å for the TF20. Electron dose was calibrated to 25 $e^-$/Å$^2$ and the defocus was set at −1.50 μm. Micrographs were recorded using a Tietz 4k × 4k TemCam-F416 CMOS camera in both cases. The Leginon automated imaging interface[68] was used for data acquisition and the Appion data processing suite[69] was applied for initial processing steps. Relion/3.0[70] was used for 2D and 3D classification steps.

**nsEMPEM—data processing**. Preliminary processing was conducted through the Appion data processing package where approximately 100,000–150,000 particles were auto-picked and extracted. Particles were then 2D-classified using Relion/3.0 into 250 classes (50 iterations), and particles with antigen-Fab qualities (roughly 50–80% of the original particles) were selected for 3D analysis. The initial 3D classification was performed using 40 classes. A low-resolution model of non-liganded HIV Env ectodomain was used as a reference for all 3D steps. Particles from similar-looking classes were then combined and reclassified, and a subgroup of 3D classes with unique structural features was further processed using 3D auto-refinement (Relion 3.0). UCSF Chimera 1.13[71] was used to visualize and segment the 3D refined maps. Finally, 3D refinement was conducted on a subgroup of particles selected after the 2D classification step (and prior to any 3D classification), and the refined model has been submitted to EMDB. Full particle stacks and 3D models used for Fab segmentation and generation of composite figures are available upon request.

**cryoEMPEM—preparation of Fab and complex samples**. We used the approach described above to generate 6–10 mg of purified polyclonal Fab samples from week 26 plasma extracted from animals Rh.32034, Rh.33104, and Rh.33311. For Rh.33172 animals, we used plasma collected at week 38. Polyclonal Fab samples were complexed with 200 μg of the corresponding BG505 SOSIP antigen used in the immunization (BG505 SOSIP MD39 for Rh.32034 and Rh.33104 samples; BG505 SOSIP.v5.2 N241/N289 for Rh.33311 sample; BG505 SOSIPv5.2(7S) N241/N289 for Rh.33172 sample) and incubated for ~18 h at room temperature. Trimer-Fab immune complexes were SEC-purified (TBS was used as the running buffer) and concentrated to 5–7 mg/ml for application onto cryoEM grids.

**cryoEMPEM—grid preparation and cryoEM imaging**. A Vitrobot mark IV (Thermo Fisher Scientific) was used for cryo-grid preparation with the four immune complex samples described above. The temperature inside the chamber was set to 10 °C, humidity was maintained at 100%, blotting time was varied within a 4–7 s range, blotting force was set to 0, and wait time was set to 10 s. For cryo-grid preparation, we used lauryl maltose neopentyl glycol (LMNG) at a final concentration of 0.005 mM. Two types of grids were used: UltrAuFoil R 1.2/1.3 (Au, 300-mesh; Quantifoil Micro Tools GmbH) and Quantifoil R 2/1 (Cu, 400-mesh; Quantifoil Micro Tools GmbH). The grids were treated with Ar/O$_2$ plasma (Solarus plasma cleaner, Gatan) for 10 s before sample application. An appropriate volume of 0.04 mM LMNG stock solution was mixed with the sample and 3 μl were immediately loaded onto the grid. After the blot step, the grids were plunge-frozen into liquid-nitrogen-cooled liquid ethane. Cryo-grids were loaded into an FEI Titan Krios electron microscope (Thermo Fisher Scientific) operating at 300 kV. The microscope is equipped with a K2 Summit direct electron detector camera (Gatan) and sample autoloader. Exposure magnification was set to 29,000 with the resulting pixel size at the specimen plane of 1.03 Å. Leginon software was used for automated data collection[68]. Specific information on the imaging of individual polyclonal complex samples can be found in Supplementary Table 3.

**cryoEMPEM—data processing**. Micrograph movie frames were aligned and dose-weighted using MotionCor2[72]. Initial data processing was performed in cryoSPARCv2.15[73]. GCTF[74] was used for estimation of CTF parameters. Immune-complex particles were picked using template picker, and two rounds of 2D classification were applied to eliminate bad particles picks and disassembled trimers. Particle stacks were transferred to Relion/3.0[70] for further processing. After one round of 2D classification in Relion, the particles belonging to classes with appropriate trimer-Fab immune complex qualities were selected and subjected to a round of 3D refinement (C3 symmetry; soft solvent mask around the trimer core). A low-pass filtered map of BG505 SOSIP trimer was used as an initial model for all 3D steps to avoid initial model bias when reconstructing maps with unknown polyclonal Fabs. 3D-aligned particles were symmetry-expanded around the C3 axis because of the trimeric nature of the antigen. This expansion collapses all epitope-paratope interfaces onto a single protomer, which simplifies the 3D classification process. However, to prevent symmetry-related copies of individual particles from aligning to themselves, particle alignment was constrained in the subsequent 3D

classification and refinement steps. 3D classification steps were performed without image alignment (--skip_align, $T = 16$) and 3D refinement steps on the selected subsets of particles were performed with local angular searches only (starting at 3.7° or 1.8° per iteration). The first round of 3D classification was run with an 80 Å sphere mask around the epitope–paratope interface and centered on the Fab-corresponding density. This is done separately for each epitope cluster where polyclonal Fabs can be detected. The number of output 3D classes was adjusted for each epitope based on the relative occupancy of bound polyclonal Fabs (typically 8–30 classes). The 80 Å sphere masks allowed for sorting on the absence/presence of Fabs at each site as well as the relative orientation. 3D classes of particles featuring structurally unique polyclonal Fabs (in terms of epitopes and orientation) were selected separately and processed independently from this point on. After a round of 3D refinement for all selected subsets of particles featuring structurally unique polyclonal Fabs, the second round of 3D classification was run (120 Å sphere mask; $K = 3$; $T = 16$). The larger mask allowed sorting based on epitope–paratope features and minimized the 3D classification problems caused by the flexible linkage between constant and variable Fab domains (mainly, reducing the influence of the relative domain positions in 3D classification and increasing the resolution of the resulting 3D classes). 3D classes of particles with the highest quality and resolution were selected and subjected to 3D refinement. Soft solvent masks around the specific trimer–Fab complex were used. A final round of 3D classification was run with a full trimer–Fab complex mask prepared in the previous step ($K = 3$; $T = 16$). The highest-resolution classes were selected for every unique trimer–Fab complex, 3D-refined and postprocessed (solvent mask around the complex; MTF correction). The postprocessed maps were used for model building and submission to EMDB. This workflow is further illustrated in Supplementary Fig. 4. Relevant data processing information (particle count at different stages, apparent map resolutions, B-factors used for map sharpening) are presented in Supplementary Table 4.

**cryoEMPEM—model building and refinement**. Postprocessed maps from Relion generated in the previous step were used for model building and refinement. The BG505 SOSIP structure from PDB entry 6vfl[15] was used as an initial model for the antigen-corresponding parts of the maps. The sequence was adjusted to match the exact BG505 SOSIP variant of the imaged polyclonal complexes (BG505 SOSIP MD39, BG505 SOSIP.v5.2 N241/N289, or BG505 SOSIP.v5.2(7S) N241/N289). Initial Fab models were generated from PDB entry 4KTE[32] by mutating all of the amino acids to alanine. The BG505 SOSIP and Fab models were docked into each map in UCSF Chimera to generate a starting model for refinement. Structural homology to published rhesus macaque antibody structures (PDB IDs: 4KTE, 4KTD, 4RFE, and 4Q2Z) was used to assign antibody heavy (H) and light (L) chains[32–34]. The principal factors for assignment were the comparisons of CDR2 and CDR3 lengths and the conformations of FR2 and FR3 regions between the H and L chains in the reconstructed cryoEM maps. Iterative rounds of manual model refinement in Coot[75] and automated model refinement in Rosetta[76] were used to refine the models into the reconstructed maps. CDR lengths of poly-alanine Fab models had to be adjusted by insertion/deletion of one or more alanine residues to match with the corresponding structural constraints imposed by the cryoEM maps. Final models were evaluated using MolProbity[77] and EMRinger[78], and the refinement statistics are shown in Supplementary Table 4. The structures were submitted to the PDB database.

**CryoEM analysis of BG505 SOSIP-T33-31 nanoparticle—grid preparation and cryoEM imaging**. BG505 SOSIP-T33-31 nanoparticle sample (in TBS) was concentrated to 4.1 mg/ml for cryo-grid application. Cryo-grids were prepared using a Vitrobot mark IV under the equivalent conditions as the polyclonal immune-complex samples described above. BG505 SOSIP-T33-31 sample, premixed with 0.005 mM LMNG, was loaded onto plasma-cleaned Quantifoil R 2/1 (Cu, 400-mesh; Quantifoil Micro Tools GmbH) grids. Following the blotting step, the grids were plunge-frozen into liquid-nitrogen-cooled liquid ethane. Cryo-grids were imaged on an FEI Talos Arctica (Thermo Fisher Scientific) microscope operating at 200 kV, equipped with a K2 Summit direct electron detector camera (Gatan) and sample autoloader. Exposure magnification was set to 36,000 and the pixel size at the specimen plane was 1.15 Å. Automated data collection was performed using the Leginon software suite[68]. Data collection information can be found in Supplementary Table 3.

**CryoEM analysis of BG505 SOSIP-T33-31 nanoparticle—data processing**. Micrograph movie frames were aligned and dose-weighted using MotionCor2[72]. All further processing steps were performed in Relion/3.0[70]. GCTF[74] was used for the estimation of CTF parameters. 233,999 particles were auto-picked and extracted. Following a round of 2D classification, 214,224 particles were selected for 3D steps. Several iterative rounds of 3D refinement and 3D classification in Relion were performed to identify a subpopulation of 110,369 particles that were used in the final 3D reconstruction of the nanoparticle core. Tetrahedral symmetry (T) was applied for the 3D steps. A soft solvent mask around the T33-31 nanoparticle core was introduced for the final 3D classification, refinement, and postprocessing steps. This solvent mask excluded the density corresponding to flexibly-linked BG505 SOSIP antigens. The final map resolution of the T33-31 nanoparticle core was 2.9 Å after postprocessing. BG505 SOSIP trimers were connected to both nanoparticle

components via flexible linkers. The flexibility prevents a joint analysis with the nanoparticle core. We applied localized reconstruction v1.2.0[79] to extract BG505 SOSIP trimer subparticles connected to T33-31A and T33-31B components (termed BG505 SOSIP(A) and BG505 SOSIP(B)). Subparticle vectors were defined by marker files generated in UCSF Chimera[71], and trimer subparticles were extracted from pre-aligned nanoparticles after the subtraction of the signal corresponding to the nanoparticle core. Each nanoparticle displayed four trimers on each component (eight total) so the number of extracted BG505 SOSIP(A) and BG505 SOSIP(B) subparticles was 441,476 (4 × 110,369). A and B subsets were processed independently in Relion. Each subparticle subset was subjected to two rounds of 2D classification and one round of 3D classification. After eliminating the cropped and low-resolution classes of subparticles, 106,478 subparticles of BG505 SOSIP(A) and 64,726 subparticles corresponding to BG505 SOSIP(B) were subjected to 3D auto-refinements with C3 symmetry. A soft solvent mask was applied for refinement and postprocessing steps. Final map resolutions were 3.7 and 3.8 Å for BG505 SOSIP(A) and BG505 SOSIP(B). Relevant data processing parameters are reported in Supplementary Table 4.

**CryoEM analysis of BG505 SOSIP-T33-31 nanoparticle—model building and refinement**. Postprocessed maps corresponding to the T33-31 nanoparticle core, BG505 SOSIP(A) and BG505 SOSIP(B) generated in the previous step were used to generate atomic models. Crystal structures of non-liganded T33-31 nanoparticle (PDB entry 4zk7) and BG505 SOSIP trimer (PDB entry 6vfl) were used as initial models for refinement of the T33-31 nanoparticle core and BG505 SOSIP(A) and (B), respectively[15,55]. Symmetry was applied for all automated refinement steps (T for the nanoparticle and C3 for the BG505 SOSIP trimers). Several rounds of Rosetta relaxed refinement[76] and manual refinement in Coot[75] were performed to generate final models. EMRinger[78] and MolProbity[77] analyses were used for model validation and generation of statistics reported in Supplementary Table 4. Refined models were submitted to the Protein Data Bank.

**Reporting summary**. Further information on research design is available in the Nature Research Reporting Summary linked to this article.

## Data availability
3D maps and models from the EM analysis have been deposited to the Electron Microscopy Databank (http://www.emdatabank.org/) and the Protein Data Bank (http://www.rcsb.org/), respectively. The accession numbers are listed in the Supplementary Table 8. In the work we have also used previously published structures from the PDB database with accession numbers 6VFL, 4KTE, 4KTD, 4RFE, 4Q2Z, and 4ZK7. Source data are provided with this paper.

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

## Acknowledgements

The authors express sincere gratitude to Darrell Irvine lab (MIT) and Novavax. Inc., for providing the SMNP and Matrix-M™ adjuvants used in the immunization experiments. The authors thank Bill Anderson, Hannah L. Turner, Charles A. Bowman and Jean-Christophe Ducom (The Scripps Research Institute) for their help with electron microscopy, data acquisition and processing. The authors also acknowledge Lauren Holden for her help on the preparation of this paper. This work was supported by grants from the National Institute of Allergy and Infectious Diseases, Center for HIV/AIDS Vaccine Immunology and Immunogen Discovery UM1AI100663 (M.C., A.B.W., D.R.B., and G.S.), Center for HIV/AIDS Vaccine Development UM1AI144462 (M.C., A.B.W., D.R.B., and G.S.), P01 AI110657 (J.P.M., R.W.S., and A.B.W.); and by the Bill and Melinda Gates Foundation and the Collaboration for AIDS Vaccine Discovery (CAVD) OPP1156262 (N.P.K. and D.B.), OPP1115782/INV-002916 (A.B.W.), OPP1132237 (J.P.M.), OPP1146996 (D.C.M.), INV-002022 (R.W.S.), and OPP1196345 (A.B.W.); and by the National Science Foundation grant DMREF 1629214 (N.P.K. and D.B.); and by the Howard Hughes Medical Institute (D.B.). Yerkes National Primate Research Center is supported by the base grant P51 OD011132. This work was also supported by the European Union's Horizon 2020 research and innovation program under grant agreement No. 681137 (M.C., R.W.S.). C.A.C. is supported by a NIH F31 Ruth L. Kirschstein Predoctoral Award AI131873 and by the Achievement Rewards for College Scientists Foundation. R.W.S. is supported by the Vici fellowship from the Netherlands Organization for Scientific Research (NWO). This work was partially funded by IAVI with the generous support of USAID, Ministry of Foreign Affairs of the Netherlands, and the Bill & Melinda Gates Foundation; a full list of IAVI donors is available at www.iavi.org. The contents of this manuscript are the responsibility of the authors and do not necessarily reflect the views of USAID or the US Government. The funders had no role in study design, data collection and analysis, decision to publish, or preparation of the paper.

## Author contributions

A.B.W., S.C., W.R.S., D.R.B., and G.S. conceived the Rhesus macaque immunization study. C.A.C., D.G.C., and A.A. helped design the immunization experiments. D.G.C., L.E.J., J.T.N., and J.B.S. executed the immunization experiments and collected serum/plasma samples. A.A. produced the BG505 SOSIP-bearing T33-31 nanoparticles and performed structural and antigenic characterization. Z.T.B. helped with processing of cryoEM data. J.C. helped produce the T33-31 nanoparticle components. C.A.C. engineered and produced the BG505 SOSIP.v5.2 N241/N289 immunogen. E.G and B.G. produced the BG505 SOSIP MD39 immunogen. J.D.A. ran site-specific glycosylation analysis on purified antigen-bearing nanoparticle components. C.A.C., L.M.S., and A.A. optimized and performed the ELISA experiments. J.B., R.B., and C.L. performed the pseudovirus neutralization experiments. C.A.C., L.M.S., and A.A. executed the nsEM-PEM experiments. L.M.S. isolated Fabs from plasma samples for all EMPEM experiments. A.A. acquired and processed the cryoEMPEM data. H.R.P., K.R., F.C., Y.R.Y., A.T.d.l.P., R.F.R., and A.A. built and refined the atomic models into the cryoEM maps. A.B.W conceptualized the study. A.B.W., G.S., S.C., W.R.S., D.R.B., M.C., and D.C.M. supervised the study and provided essential guidance. D.B., N.P.K., R.W.S., and J.P.M. gave critical feedback regarding study design and data interpretation. A.B.W. and A.A. wrote the original draft of the paper. All authors contributed to the manuscript text by assisting in writing and/or providing critical feedback.

## Competing interests

The authors declare no competing interests.
