## [Peer Review File · Nature Communications]

REVIEWER COMMENTS

Reviewer #1 (Remarks to the Author):

The authors present a cryo-EM refinement and classification-based strategy (cryoEMPEM) to map the epitopes of polyclonal antibodies. They applied this strategy to evaluate several BG505 Env trimer vaccines, including purified soluble Env trimers and nanoparticle presented Env trimers. Using cryoEMPEM, they were able to obtain high resolution antibody-antigen complex structures. Based on these structures, they performed detailed analyses of epitope-paratope interfaces and compared their polyclonal antibody responses with previous immunization studies. Besides, they also used traditional methods such as ELISA and TZM-bl pseudovirus infection assay to evaluate the antigen-specific responses and neutralization titers. These results allowed them to understand the immunogenicity of the vaccines and provide essential information for engineering the next generation of vaccines.

The methods part of the paper is impressive. cryoEMPEM seems to be a really good strategy to quickly map the epitopes. Compared with traditional methods such as monoclonal antibody isolation, or neutralization assays using large panel of HIV strains, this EM based strategy seems easier and more straight forward. Compared with nsEMPEM, cryoEMPEM gives higher resolution information that enables molecular-level understanding of polyclonal antibody responses. In the Material and Methods part, the authors provided enough information about sample preparation and cryoEM data processing procedures. As someone with cryoEM experience, I think the Methods section is well written and should allow others to follow cryoEMPEM.

The immunization part, on the other hand, might not be very exciting. Neither the BG505 SOSIP based vaccines nor the nanoparticles induced promising broadly neutralizing antibodies. But the authors still carefully characterized the antibody responses, especially their epitopes and paratopes. Their analyses revealed specific limitations of the vaccines and will provide good references for future HIV vaccine design efforts.

Overall the manuscript will be insightful for the field and I support this publication. But I hope the authors could address the following questions:

1. In figure 1B, C the authors showed and compared the binding and neutralization titers of two groups. Can authors also compare these titers with previous immunization experiments using other BG505 SOSIP based vaccines and comment on this in the results section?
2. The new gp120-gp120 interface epitope looks very interesting. I'm assuming these antibodies are not observed in previous immunization studies. If so, can the authors explain why the mutations of BG505 SOSIP .v5.2 N241/N289 can lead to these antibodies? The results in SI figure 1 do indicate that these stabilizing mutations are required for gp12-gp120 interface antibodies binding but I don't quite understand the mechanism behind. Besides, will all gp120-gp120 interface antibodies sterically prevent CD4 binding? Can the authors conclude this just based on negative stain structures or it will require cryoEM resolution? It will be nice if the authors could include several coordinates in their rebuttal.
3. During cryoEMPEM, I'm surprised that the authors increased the size of the spherical mask around the Fab in the 2nd round of 3D classification, as I would imagine a smaller or at least the same size mask will give higher map quality within the mask. Can the authors explain their rationale? In the methods section the authors claimed that "The larger mask allowed sorting based on the epitope-paratope features" but I don't quite understand this

sentence.

4. Antibodies targeting the C3/V5 epitope are the primary contributors to the autologous neutralization in this paper, and also in many other immunization studies using nonhuman primates. However, I don't know any patient derived bNAb targeting this epitope. Is it because these NAbs are strain specific? Or it's an animal model bias? Can the authors comment on this?

5. The C54, C73, C74, and C561 disulfide bond network looks very interesting. Based on the current resolution, does the new network only involve side chain conformational changes or it also involves backbone changes as well? Besides, do authors only see this for Rh 33311-pAbc-3 structure or for other Rh 33311 structures as well?

6. All group 3 animals developed V1/V2/V3 antibodies. This is as expected based on the nanoparticle design and also consistent with previous nanoparticle immunization studies. However, almost all group 3 animals developed antibodies targeting the base epitope. Do the authors have a good explanation for this? In the discussion session the authors suggest it could be due to nanoparticle disassemble. Is there any evidence for that?

7. In Figure 6D, the authors should also indicate the neutralization titers like they did in figure 1D.

8. In the discussion session the authors claimed that cryoEMPEM analysis can be completed within ~10 days from serum/plasma collection. I believe it is possible for the authors as they are familiar with the procedures. However, I really doubt even a cryoEM expert with enough resources can achieve this at the beginning. Therefore I hope the authors can give a more detailed and more practical time estimation. This will give other users a better idea of what to expect when they choose to perform cryoEMPEM themselves.

9. Personally, I think both nsEMPEM and cryoEMPEM are very promising strategies for quick characterization of antibody responses. This is beyond the scope of this paper but I hope the authors can write a protocol/methods paper to describe the detailed procedures, including potential problems and how to trouble shoot. I look forward to seeing the potential of ns/cryoEMPEM, not only for vaccine evaluation, but also for protein engineering and directed evolution, etc.

Reviewer #2 (Remarks to the Author):

Antanasijevic and colleagues developed a high-resolution structure-determination workflow by cryo-EM (cryo-EM-based polyclonal epitope mapping) to discover polyclonal antibody responses to HIV Env immunogens. Upon immunization of three groups of rhesus macaques with two different soluble or a nanoparticle BG505 SOSIP-based immunogen, the authors isolated polyclonal antibodies and determined in total 21 high-resolution cryo-EM reconstructions of HIV trimer-immune-fab complexes. It is exciting to see how the elegant and efficient cryo-EM processing workflow revealed high-resolution epitope information for these heterogenous polyclonal antibody complexes. The cryo-EM statistics look great and the shown density maps confirm the high quality of the presented data. The impressive number of determined structures provide a basis for the design HIV vaccines based on BG505 SOSIP immunogens and highlights the importance of carefully introducing stabilizing mutations while maintaining neutralizing responses. Given the importance of HIV infections in public health, this study has very likely a high impact for the design of future vaccines. Moreover, the developed structure determination workflow is not limited to HIV and is applicable for studying different vaccine platforms in a broad range of diseases. Overall, I find that this work is of high quality and of high significance for a broad readership. My comments below are minor and refer mainly to data presentation.

Comments:

1. The cryoEMPEM workflow is one of the key points of this work. I would find it very helpful if the authors can briefly explain the key steps of the purification and processing workflow in the main text (not only in the material and methods).
2. I would find it also very helpful if the authors could illustrate key steps of the cryoEMPEM processing workflow in main Figure 2 (currently only in Figure S2) and provide the color code legend for the respective antibody binding sites (as shown in Figure 1) also in Figure 2
3. For the cryoEMPEM workflow, it would be great if the authors can show a SEC profile of the final purification before grid freezing (highlighting also the selected fractions) to understand the homogeneity/heterogeneity of the sample.
4. For Figure 1A/Table S1, it could be helpful if the authors can also highlight the position of the stabilizing or immunogenic mutations used in this study on a structural model.
5. For Figure 3 + 4 it is difficult to see the glycans and fab positions on the overview panel of the antigen model. It may be easier to appreciate the position of the Fabs and glycans if the antigen is shown as a space-filling/ surface representation instead of the ribbon/cartoon style.
6. For Figure S1+S7, it is almost impossible to read the scale on the micrographs. A descriptive explanation in the legend could be helpful

Reviewer #3 (Remarks to the Author):

Antanasijevic et al. used several stabilized BG505 SOSIP constructs to immunize the monkeys and analyzed the profile of polyclonal antibody responses by using (cryoEMPEM). This study provided very useful information BG505 immunigen and this manuscript was well organized. I suggest revise before accepting.

- 1.The animals had been immunized four times, but the interval between two doses is different. Please describe the reason.
- 2.In Figure 1B, the titre of binding antibody in sera from monkey immunized with BG505 SOSIP MD39 had increased with more times of immunization, but the titre of binding antibody in monkey immunized with BG505 SOSIP.v5.2 N241/N289 had decreased, and that in monkey immunized with BG505 SOSIP-T33-31(Figure 6B) had no change with more times of immunization. Please describe the reason. Figure 6B and C had no control and authors described in text that results in Figure 6B and C were compared with those in Figure 1 B and C, but didn't mention whether the tests for Figure 6B,C and Figure 1B, C had been done in same time.
- 3.Figure 1 D showed the polyclonal antibody response in monkey after third immunization. But Figure 1 C showed that the titre of neutralizing antibody had increased with more times of immunization. Why didn't select sera with highest titer of neutralizing antibody for analyzing nsEMPEM.
- 4.In Figure 1 D, BG505 SOSIP MD39 was used to immunize 6 monkeys. Among them, two produced neutralizing antibody against N241 GH, and two produced neutralizing antibody against N289 GH(N355) . Among 6 monkeys immunized with BG505 SOSIP.v5.2 N241/N289, only 1 monkey produced neutralizing antibody against N289 GH(N355). This result cannot reach the conclusion that "the presence of N241 and N289 glycans in the BG505 SOSIP.v5.2 N241/N289 immunogen suppressed the antibody response against this epitope". More tests needed to confirm this.
- 5.Didn't show whether the nanoparticle could increase the titre of neutralizing antibody.

REVIEWER COMMENTS

Reviewer #1 (Remarks to the Author):

The authors present a cryo-EM refinement and classification-based strategy (cryoEMPEM) to map the epitopes of polyclonal antibodies. They applied this strategy to evaluate several BG505 Env trimer vaccines, including purified soluble Env trimers and nanoparticle presented Env trimers. Using cryoEMPEM, they were able to obtain high resolution antibody-antigen complex structures. Based on these structures, they performed detailed analyses of epitope-paratope interfaces and compared their polyclonal antibody responses with previous immunization studies. Besides, they also used traditional methods such as ELISA and TZM-bl pseudovirus infection assay to evaluate the antigen-specific responses and neutralization titers. These results allowed them to understand the immunogenicity of the vaccines and provide essential information for engineering the next generation of vaccines.

The methods part of the paper is impressive. cryoEMPEM seems to be a really good strategy to quickly map the epitopes. Compared with traditional methods such as monoclonal antibody isolation, or neutralization assays using large panel of HIV strains, this EM based strategy seems easier and more straight forward. Compared with nsEMPEM, cryoEMPEM gives higher resolution information that enables molecular-level understanding of polyclonal antibody responses. In the Material and Methods part, the authors provided enough information about sample preparation and cryoEM data processing procedures. As someone with cryoEM experience, I think the Methods section is well written and should allow others to follow cryoEMPEM.

The immunization part, on the other hand, might not be very exciting. Neither the BG505 SOSIP based vaccines nor the nanoparticles induced promising broadly neutralizing antibodies. But the authors still carefully characterized the antibody responses, especially their epitopes and paratopes. Their analyses revealed specific limitations of the vaccines and will provide good references for future HIV vaccine design efforts.

Overall the manuscript will be insightful for the field and I support this publication. But I hope the authors could address the following questions:

1. In figure 1B, C the authors showed and compared the binding and neutralization titers of two groups. Can authors also compare these titers with previous immunization experiments using other BG505 SOSIP based vaccines and comment on this in the results section?

Per reviewer's suggestion, the comparison to previously published immunization experiments has been added the results section (lines 107-111).

2. The new gp120-gp120 interface epitope looks very interesting. I'm assuming these antibodies are not observed in previous immunization studies. If so, can the authors explain why the mutations of BG505 SOSIP .v5.2 N241/N289 can lead to these antibodies? The results in SI figure 1 do indicate that these stabilizing mutations are required for gp12-gp120 interface antibodies binding but I don't quite understand the mechanism behind. Besides, will all gp120-gp120 interface antibodies sterically prevent CD4 binding? Can the authors conclude this just based on negative stain structures or it will require cryoEM resolution? It will be nice if the authors could include several coordinates in their rebuttal.

To address the reviewer's questions, we have included the following edits to the revised version of the manuscript:

- a) We discuss the impact of stabilizing mutations in SOSIP.v5.2 N241/N289 on elicitation of antibody response against the C1 and V3 epitopes (lines 505-514 in the Discussion section)
- b) We have performed additional analysis of antibodies targeting the CD4bs, silent face and gp120-gp120 interface epitopes. These data are shown in Supplementary Figure 8D, and discussed in the Results section (lines 326-332)

3. During cryoEMPEM, I'm surprised that the authors increased the size of the spherical mask around the Fab in the 2nd round of 3D classification, as I would imagine a smaller or at least the same size mask will give higher map quality within the mask. Can the authors explain their rationale? In the methods section the authors claimed that "The larger mask allowed sorting based on the epitope-paratope features" but I don't quite understand this sentence.

There were 3 main reasons for expanding the spherical mask used in the 2nd round of 3D classifications. First, we observed that repeating 3D classifications with the smaller 80Å sphere mask lead to 3D classes that differed solely in the conformation of the constant domain of the Fab (this domain constitutes ~50% of the Fab mass). Given our main focus on the variable Fab domain and the epitope-paratope contacts, this type of 3D classification is not optimal. We empirically observed that by applying a larger ~120Å mask, Relion's 3D classification algorithm puts less weight on the conformation of the constant Fab domain. Second, we observed that the estimated map resolutions in 3D classes were relatively low (~6-10Å range) when using a smaller ~80Å mask. This is likely due to the small size of the Fab (~50kDa) and the flexibility that exists between the constant and the variable regions of the Fab. Consequently, this type of 3D classification was not optimal for removing heterogeneity that exists at the high-resolution level and separating classes that appear similar at low resolution ("data polishing"). Including the epitope within the mask resulted in 3D classes with higher resolution estimates (up to ~4Å). Third, in certain cases it may be beneficial to focus on the epitope features (e.g. flexible loop conformations, glycan conformations and occupancy) during 3D classification as oppose to just focusing on the Fab. In this study, we did not observe any major differences in epitopes features, but from our work with monoclonal antibody complexes we know that these effects can occur.

In the Methods section of the revised manuscript we have included a more detailed explanation of our rationale (lines: 799-802).

4. Antibodies targeting the C3/V5 epitope are the primary contributors to the autologous neutralization in this paper, and also in many other immunization studies using nonhuman primates. However, I don't know any patient derived bNAb targeting this epitope. Is it because these NABs are strain specific? Or it's an animal model bias? Can the authors comment on this?

The C3/V5 epitope is proximal to the CD4bs and interference with CD4 binding is the most likely mechanism by which antibodies targeting this site can neutralize the BG505 virus. Neutralizing antibodies targeting this epitope have been elicited by BG505 SOSIP in several different animal models (macaques, rabbits, guinea pigs and mice). Other vaccine-elicited antibodies contributing to autologous neutralization in BG505 target the N241/N289 glycan hole, V1/V2/V3 and the fusion peptide epitopes, but they are typically subdominant compared to C3/V5 NABs. However, the sequence of the V5 loop is highly variable (in terms of aa composition, length and glycosylation), therefore these antibodies do not cross-react with other HIV Env strains.

To address the reviewer's questions, we have now included additional clarification in the Results section (lines: 190-195)

5. The C54, C73, C74, and C561 disulfide bond network looks very interesting. Based on the current

resolution, does the new network only involve side chain conformational changes or it also involves backbone changes as well? Besides, do authors only see this for Rh 33311-pAbc-3 structure or for other Rh 33311 structures as well?

Based on the available structural data (and at the current map resolutions), we can definitely confirm that the change from “intended” C54-C74/C73-C561 to “cross-linked” C54-C73/C74-C561 coupling does require multiple conformational changes on the level of both, side chain and main chain. Residue C54 only alters the conformation of the cysteine side chain while the main chain does not appear to move significantly. For the C73 and C74 residues in the C1 loop, the two residues need to rotate by $\sim 180^\circ$ around the main chain, which alters the polypeptide backbone in the area and induces the reorganization of nearby residues 68-72 to a different conformation. Finally, the HR1 region around the residue C561 needs to undergo extensive restructuring for the switch to take place. In the “cross-linked” state the surrounding residues (P559-K567) form a short helix, but the helical conformation would not be supported if the “intended” disulfide coupling occurred.

“Cross-linked” network has been observed in all of Rh.33311 pAbC maps, including the highest resolution maps (pAbC-1 and pAbC-5). However, the map quality is the lowest in HR1 and C1 regions surrounding the residues in question, which suggests high degree of conformational heterogeneity. It is likely that both states exist in BG505 SOSIP.v5.2 and that we are reconstructing an average that is enriched for the “cross-linked” population of particles. In fact, different monomers within a trimer can be coupled differently with respect to this disulfide network.

To address the reviewer’s questions, we have now included additional figure (Supplementary Figure 8C) and data discussion in the Results section (lines: 303-310)

6. All group 3 animals developed V1/V2/V3 antibodies. This is as expected based on the nanoparticle design and also consistent with previous nanoparticle immunization studies. However, almost all group 3 animals developed antibodies targeting the base epitope. Do the authors have a good explanation for this? In the discussion session the authors suggest it could be due to nanoparticle disassembly. Is there any evidence for that?

We propose that nanoparticle disassembly is a contributing factor in elicitation of base-directed antibodies because of 2 main reasons. (1) Biolayer interferometry analysis demonstrates that base-specific antibodies (RM19R and RM20A3) and antibodies targeting base-proximal epitopes (35O22 and 3BC315) can readily interact with free nanoparticle components (BG505 SOSIP T33-31A and BG505 SOSIP T33-31B) but the binding is significantly reduced to fully assembled nanoparticles (BG505 SOSIP T33-31), due to antigen crowding on the nanoparticle surface. (2) Base-directed antibodies reconstructed using nsEMPEM (Figure 6) and cryoEMPEM (Figure 7), engage their epitopes at a very steep angle of approach. It is highly unlikely that such steep antibodies would be elicited by fully assembled nanoparticle.

However, due the lack of more definitive data to assess disassembly *in vivo* we decided to be conservative with our discussion of these data.

7. In Figure 6D, the authors should also indicate the neutralization titers like they did in figure 1D.

Figure 6D has now been updated with neutralization titers (ID_{50}) per animal for each time point as done in Figure 1D.

8. In the discussion session the authors claimed that cryoEMPEM analysis can be completed within ~ 10 days from serum/plasma collection. I believe it is possible for the authors as they are familiar with the procedures. However, I really doubt even a cryoEM expert with enough resources can achieve this at the beginning. Therefore I hope the authors can give a more detailed and more practical time estimation. This

will give other users a better idea of what to expect when they choose to perform cryoEMPEM themselves.

To address the reviewer's comment, we have now updated the discussion section with a more detailed analysis of individual steps within cryoEMPEM workflow and time estimates for each step (lines 471-476)

9. Personally, I think both nsEMPEM and cryoEMPEM are very promising strategies for quick characterization of antibody responses. This is beyond the scope of this paper but I hope the authors can write a protocol/methods paper to describe the detailed procedures, including potential problems and how to trouble shoot. I look forward to seeing the potential of ns/cryoEMPEM, not only for vaccine evaluation, but also for protein engineering and directed evolution, etc.

We sincerely appreciate the support and enthusiasm from the reviewer, as ns/cryoEMPEM methods represent something that we want to pursue long-term in the lab. Coincidentally, we are in the process of preparing a protocol/methods paper on novel optimized strategies for EMPEM analysis (focusing primarily on polyclonal sample processing and expanding the detection range).

Reviewer #2 (Remarks to the Author):

Antanasijevic and colleagues developed a high-resolution structure-determination workflow by cryo-EM (cryo-EM-based polyclonal epitope mapping) to discover polyclonal antibody responses to HIV Env immunogens. Upon immunization of three groups of rhesus macaques with two different soluble or a nanoparticle BG505 SOSIP-based immunogen, the authors isolated polyclonal antibodies and determined in total 21 high-resolution cryo-EM reconstructions of HIV trimer-immune-fab complexes. It is exciting to see how the elegant and efficient cryo-EM processing workflow revealed high-resolution epitope information for these heterogenous polyclonal antibody complexes. The cryo-EM statistics look great and the shown density maps confirm the high quality of the presented data. The impressive number of determined structures provide a basis for the design HIV vaccines based on BG505 SOSIP immunogens and highlights the importance of carefully introducing stabilizing mutations while maintaining neutralizing responses. Given the importance of HIV infections in public health, this study has very likely a high impact for the design of future vaccines. Moreover, the developed structure determination workflow is not limited to HIV and is applicable for studying different vaccine platforms in a broad range of diseases. Overall, I find that this work is of high quality and of high significance for a broad readership. My comments below are minor and refer mainly to data presentation.

Comments:

1. The cryoEMPEM workflow is one of the key points of this work. I would find it very helpful if the authors can briefly explain the key steps of the purification and processing workflow in the main text (not only in the material and methods).

To address the reviewer's comment, we have now included additional description of the sample preparation and data processing workflows in the Results section (lines 138-150).

2. I would find it also very helpful if the authors could illustrate key steps of the cryoEMPEM processing workflow in main Figure 2 (currently only in Figure S2) and provide the color code legend for the respective antibody binding sites (as shown in Figure 1) also in Figure 2

To address the reviewer's comment, we have updated Figure 2 in the revised version of the manuscript. An illustration of the data processing workflow is now added as Figure 2A. The color-code legend for different epitopes is included in updated Figure 2B.

3. For the cryoEMPEM workflow, it would be great if the authors can show a SEC profile of the final purification before grid freezing (highlighting also the selected fractions) to understand the homogeneity/heterogeneity of the sample.

To address the reviewer's comment, we have now included a Supplementary Figure 3 in the manuscript with the SEC profiles for different immune complexes.

4. For Figure 1A/Table S1, it could be helpful if the authors can also highlight the position of the stabilizing or immunogenic mutations used in this study on a structural model.

Per reviewer's comment, we have now included a new Supplementary Figure 1 with the stabilizing mutations highlighted on BG505 SOSIP models.

5. For Figure 3 + 4 it is difficult to see the glycans and fab positions on the overview panel of the antigen model. It may be easier to appreciate the position of the Fabs and glycans if the antigen is shown as a space-filling/ surface representation instead of the ribbon/cartoon style.

The proposed edits have been incorporated into the revised Figures 3 and 4. The overview panels featuring full immune complexes are now illustrated using surface representation as opposed to ribbon.

6. For Figure S1+S7, it is almost impossible to read the scale on the micrographs. A descriptive explanation in the legend could be helpful

To address the reviewer's comment, we have updated the captions of the two Supplementary Figures with the size/scale information.

Reviewer #3 (Remarks to the Author):

Antanasijevic et al. used several stabilized BG505 SOSIP constructs to immunize the monkeys and analyzed the profile of polyclonal antibody responses by using (cryoEMPEM). This study provided very useful information BG505 immunogen and this manuscript was well organized. I suggest revise before accepting.

1.The animals had been immunized four times, but the interval between two doses is different. Please describe the reason.

The immunization regimen has been adjusted to match the previous research performed by our groups (Pauthner et al., 2017) and make appropriate comparisons to the data that was published in that study and as a follow-up of that study (Pauthner et al., 2019; Nogal et al., 2020; Zhao et al., 2020). In the text we reference these manuscripts and compare our results.

The original immunogen regimen was extended by additional, 4th immunogen dose (week 36). This was done to boost the levels of antibodies against epitopes that are weakly immunogenic (particularly the neutralizing antibody epitopes).

2.In Figure 1B, the titre of binding antibody in sera from monkey immunized with BG505 SOSIP MD39 had

increased with more times of immunization, but the titre of binding antibody in monkey immunized with BG505 SOSIP.v5.2 N241/N289 had decreased, and that in monkey immunized with BG505 SOSIP-T33-31 (Figure 6B) had no change with more times of immunization. Please describe the reason. Figure 6B and C had no control and authors described in text that results in Figure 6B and C were compared with those in Figure 1 B and C, but didn't mention whether the tests for Figure 6B,C and Figure 1B, C had been done in same time.

The peak binding antibody titers were measured after each immunogen dose. Overall (across all 3 groups), the trend is that peak binding titers (EC_{50}) increase significantly after the 1st and 2nd immunogen dose and subsequently plateau in the 10^2 – 10^3 range (Supplementary Table 1). The exception is week 10 data for Grp 2 (following the 2nd immunogen dose) where three animals displayed EC_{50} titers $>10^3$. For this group of animals, the mean titers are ~3-fold higher at week 10 compared to weeks 26 and 38. While this effect is significant, we do not believe it was caused by the specific experimental conditions (i.e., immunogen, adjuvant, regimen etc.). Similar observations (i.e., the plateauing or even, decrease, of antibody binding titers after additional immunizations) has been observed in the past with rhesus macaques and BG505 SOSIP immunogens (Sanders et al, 2015). Presently, we do not have an exact explanation for this difference in the kinetics of the immune response, and we speculate that animal-to-animal variability may have influenced the results. In general, there is a fairly large amount of variability across NHP immunizations in the HIV field that are difficult to explain. Importantly, the binding antibody titers do not appear to have an effect on epitope targeting, as supported by the overlap between the epitopes we characterized in this paper, and previously published studies (Dingens et al., 2021; Cottrell et al., 2020; Nogal et al., 2020; Zhao et al., 2020). Therefore, we have high confidence that these variations in binding titers did not affect our structural analysis by cryoEMPEM - the main focus of this work.

All of the immunization experiments were performed at the same time and under the same experimental conditions. Therefore, the data is fully comparable between different groups of animals. In the revised manuscript we have included additional clarifications regarding this point (lines 386-388).

3. Figure 1 D showed the polyclonal antibody response in monkey after third immunization. But Figure 1 C showed that the titre of neutralizing antibody had increased with more times of immunization. Why didn't select sera with highest titer of neutralizing antibody for analyzing nsEMPEM.

We selected week 26 samples for this analysis because of a very practical reason. These immunization experiments are a basis of several different studies to be published in the near future. As a consequence of material sharing between multiple groups there was insufficient week 38 serum (specifically for Grp 1) for all nsEMPEM analyses and for the follow-up cryoEMPEM analyses. Therefore, we decided to use week 26 serum.

Importantly, the comparison of nsEMPEM data for week 26 and week 38 time point in Grp 3, shows very little qualitative difference in detected antibody specificities. Therefore, we believe that week 26 data is representative and while the additional immunization at week 38 may increase antibody levels against different sites, it is unlikely to yield new epitope specificities on a per-group level.

4. In Figure 1 D, BG505 SOSIP MD39 was used to immunize 6 monkeys. Among them, two produced neutralizing antibody against N241 GH, and two produced neutralizing antibody against N289 GH(N355). Among 6 monkeys immunized with BG505 SOSIP.v5.2 N241/N289, only 1 monkey produced neutralizing antibody against N289 GH(N355). This result cannot reach the conclusion that "the presence of N241 and N289 glycans in the BG505 SOSIP.v5.2 N241/N289 immunogen suppressed the antibody response against this epitope". More tests needed to confirm this.

We partially based our conclusion on the fact that N241/N289 glycans were also present in the nanoparticle immunogen (Grp 3) and none of the animals from that group developed glycan hole responses). 4/6 animals from Grp 1 (~67%) immunized with BG505 SOSIP MD39, that lacks the two glycans, had detectable GH response. On the other hand, 1/11 (~9%) of the animals from Grps 2 and 3, that received BG505 SOSIP with N241 and N289 glycans present, developed this response. We believe that this data supports our conclusion. Furthermore, this finding is consistent with previous immunization experiments with different BG505 SOSIP glycoforms in rabbit animal model (Ringe et al., 2019).

We have now included additional clarification regarding this point in the Results section (lines 124-126).

5. Didn't show whether the nanoparticle could increase the titre of neutralizing antibody.

The nanoparticle immunogen (used for immunizing Grp 3 animals) resulted in decreased neutralizing antibody titers compared to the two BG505 SOSIP trimer immunogens (Grps 1 and 2). These lower antibody titers to BG505 are likely contributed by the "distracting" immune response to the nanoparticle itself. The autologous neutralization data on a per animal basis is presented in revised Figure 6 and in Supplementary Table 2.

REVIEWER COMMENTS

Reviewer #1 (Remarks to the Author):

The authors have answered all my questions in the 1st round review very well. Also they have modified their manuscript accordingly to address these questions. I support the publication and hope the results will be insightful for the HIV vaccine and structural biology field.

Reviewer #2 (Remarks to the Author):

The authors addressed all of my points and concerns and provided additional data, illustrations and explanations that greatly improved this manuscript.

I do not have any further comments and fully support publication of this work.

Reviewer #3 (Remarks to the Author):

I am satisfied with the responses and have no comments further.